



# Computational modelling and analytical validation of singular geometric effects in fault data using a combinatorial algorithm

Michał P. Michalak[1], Janusz Morawiec[2], Peter Menzel[3]

[1]Faculty of Geology, Geophysics and Environmental Protection, AGH University of Krakow, Mickiewicza 30, 30-059 Cracow, Poland, ORCID: https://orcid.org/0000-0002-1376-235X
[2]Institute of Mathematics, Faculty of Science and Technology, University of Silesia in Katowice, Bankowa 14, 40-007 Katowice, Poland, ORCID: https://orcid.org/0000-0002-0310-867X
[3]Institute of Geophysics and Geoinformatics, TU Bergakademie Freiberg, Gustav-Zeuner-Straße 12, 09599 Freiberg, Germany

*Correspondence to*: Michał P. Michalak (michalm@agh.edu.pl )

**Abstract.**

This study analyzes the directional properties of geological faults using triangulations to model displaced horizons. We investigate two scenarios: one without elevation uncertainties and one with such uncertainties. Through formal mathematical proofs and computational experiments, we explore how triangular surface data can reveal geometric characteristics of faults. Our formal analysis introduces four propositions of increasing generality, demonstrating that in the absence of elevation errors, duplicate elevation values lead to identical dip directions. For the scenario with elevation uncertainties, we find that the expected dip direction remains consistent with the error-free case. These findings are further supported by computational experiments using a combinatorial algorithm that generates all possible three-element subsets from a given set of points. The results offer insights into predicting fault geometry in data-sparse environments and provide a framework for analyzing directional data in topographic grids with imprecise elevation data. This work has significant implications for improving fault modeling in geological studies, particularly when dealing with limited or uncertain data.





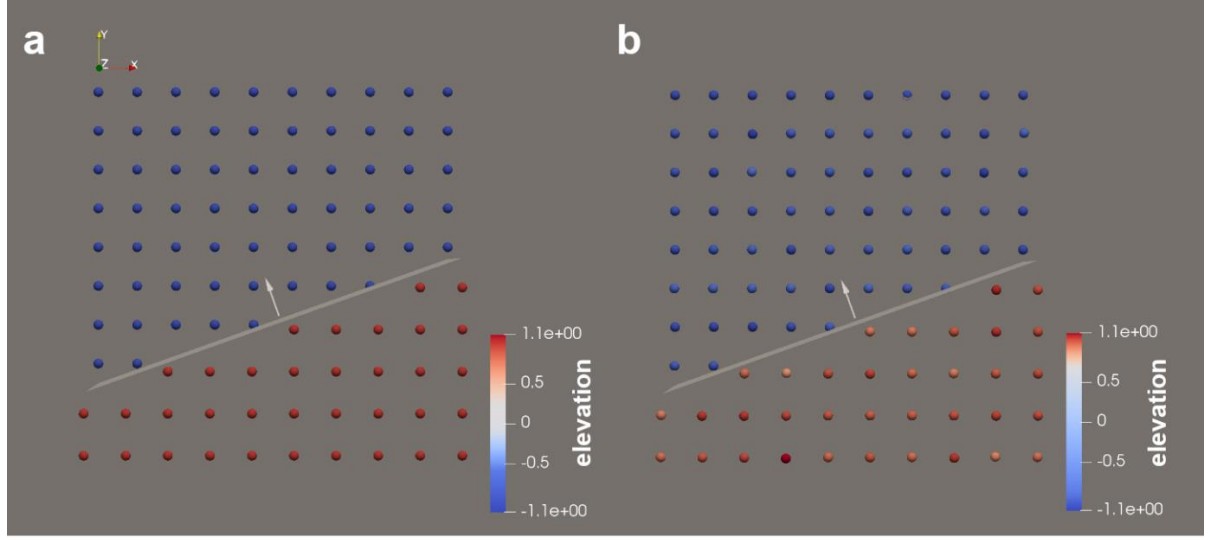

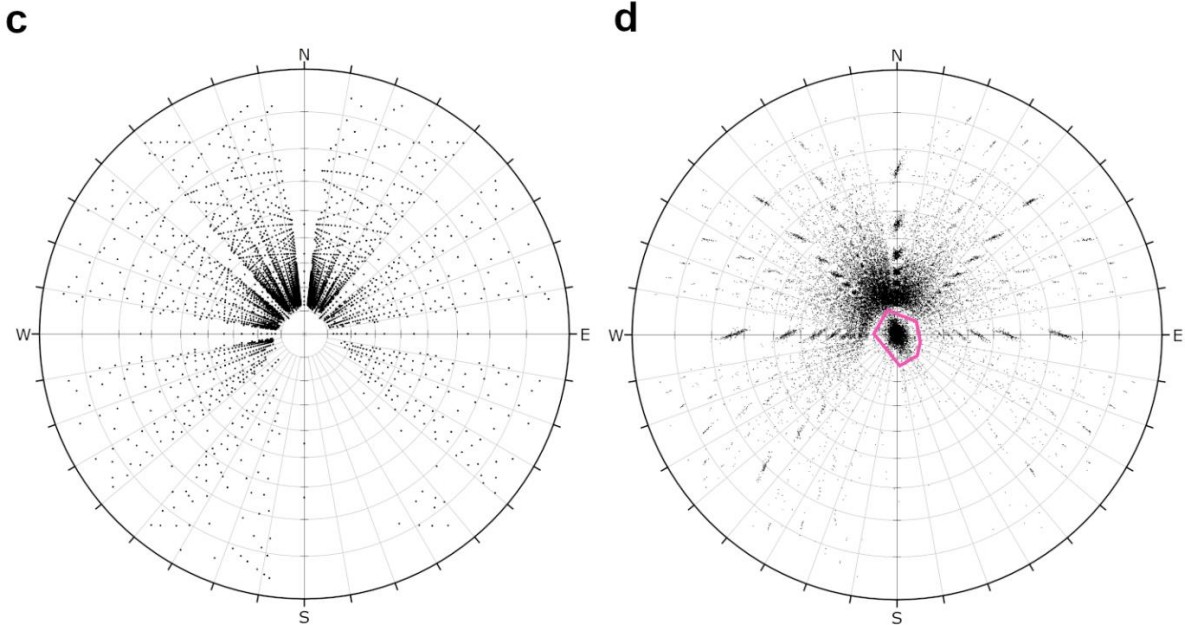



## 1 Introduction

Faults influence numerous practical aspects of subsurface geology, including groundwater flow (Bense et al., 2013), hydrocarbon entrapments (Aydin, 2000), and localized mineralization (Person et al., 2008). In areas with sparse geological data, such as subsurface regions between boreholes, inferring the geometric properties of faults presents significant challenges (Lark et al., 2013; Godefroy et al., 2019). Sparse environments often contain large epistemic uncertainty—uncertainty arising from a lack of knowledge—which can complicate the interpretation of geological structures (Bowden, 2004). While collecting more data can reduce uncertainty (Bond, 2015; Dowd, 2018), practical constraints often make this infeasible.

Recent studies have attempted to reduce epistemic uncertainty in structural geology using triangulations and combinatorial algorithms to analyze fault geometry (Michalak et al., 2021). For example, Michalak et al. demonstrated that triangles sampled from the walls (see terminology in Fig. 1) of a fault can exhibit counterintuitive behaviors, such as reversed dip directions (towards the upper wall – see Fig. 4b) and identical dip directions from different triangles. This raises intriguing questions about the geometric behavior of triangulated models under sparse data conditions.

This paper builds on this work by providing formal mathematical evidence that a combinatorial algorithm can reduce epistemic uncertainty in sparse environments. We propose a robust framework for predicting fault geometry in data-limited scenarios. To support this, we present formal analyses of two scenarios: one with perfect elevation data (Proposition 1, Proposition 2) and one accounting for elevation uncertainties (Proposition 3, Proposition 4), and explore the statistical implications of our findings using directional data. Following the formal analysis, we demonstrate the consequences of these theoretical results in the analysis of 2D and 3D (Fig. 2) directional data derived from topographic grids, which typically consist of points with approximate elevations—commonly observed in bathymetric data sets (Gridded Bathymetry Data, 2024).

## 2 Background

Uncertainty in geological modeling is a widespread issue affecting various aspects of subsurface analysis. These uncertainties stem from incomplete data, particularly in sparse environments with limited borehole data or surface observations. A key challenge in such cases is accurately relating parts of the study area to lithological units or other geological structures (Wellmann and Regenauer-Lieb, 2012). For example, uncertainty can arise from errors in borehole paths (Pakyuz-Charrier et al., 2018) or in the resulting geological models themselves (Pakyuz-Charrier et al., 2019; Liang et al., 2021). To manage these uncertainties, several methods have been developed, including uncertainty propagation techniques, such as the





Markov Chain Monte Carlo (MCMC) method, which estimates uncertainty by feeding model generators with probabilistic input data (De La Varga and Wellmann, 2016; Pakyuz-Charrier et al., 2019).

In sparse geological settings, combinatorial algorithms have emerged as a valuable tool for interpreting fault-related data. One notable example is Godefroy et al. (Godefroy et al., 2019), who developed a method to partition sparse fault evidence into fault clusters using combinatorial techniques. The authors demonstrated that this approach could handle sparse data, but the computational cost increases rapidly as data size grows, governed by Stirling numbers (Allenby and Slomson, 2010).

    Orientation measurements, such as dip and dip direction, are critical in subsurface geological modeling and are
traditionally collected through fieldwork or outcrop analogs (La Fontaine et al., 2021). These measurements serve as input for co-kriging methods, which combine point data with orientation information to model subsurface structures  (de la Varga et al., 2018). More recently, triangulated data has been widely used in geological modeling to represent 3D surfaces (Merland et al., 2014; Collon et al., 2015; Aydin and Caers, 2017). Triangulations, created by connecting points sampled from geological surfaces, allow for the analysis of orientation data by calculating the normal vectors of triangles. This approach is valuable in
detecting geological features like faults by clustering these vectors (Michalak et al., 2022).

    Despite its utility, triangulation-based analysis faces limitations, particularly when dealing with sparse data environments where the number of triangles available for analysis is reduced. The use of combinatorial algorithms offers a promising alternative by generating all possible triangle configurations from a given data set (Michalak et al., 2021).

**3 Methodology**

**3.1 Lipski algorithm**

We applied the combinatorial Lipski algorithm to reduce epistemic uncertainty in determining fault orientations. This algorithm generates all possible three-element subsets (triangles) from a given set of boreholes (Lipski, 2004). This approach systematically creates every possible triangle configuration, enabling a comprehensive geometric analysis. We generated all
$k$-element subsets ($k$=3) from an $n$-element set (where $n$ is the total number of borehole locations) to estimate the fault





orientation. The algorithm's efficiency allowed us to handle large datasets while ensuring that all potential fault-related triangles were analyzed. The full description of the Lipski algorithm can be found in Appendix A.

## 3.2 Singular geometric effects

Building on the results from Michalak et al. (Michalak et al., 2021), we extended the analysis of fault-related triangles to
account for the observed phenomenon where approximately 8% of triangles exhibited reversed dip directions (Fig. 4b). We hypothesize that this behavior is controlled by two main factors: the orientation of the edge lying on the horizontal part of the fault (hanging or footwall) and the position of the third point relative to this edge (Proposition 1, Fig. 3a). Using formal proofs (see Appendix B), we validated this hypothesis and demonstrated that the dip direction depends on whether the third point lies to the left or right of the fault's edge.

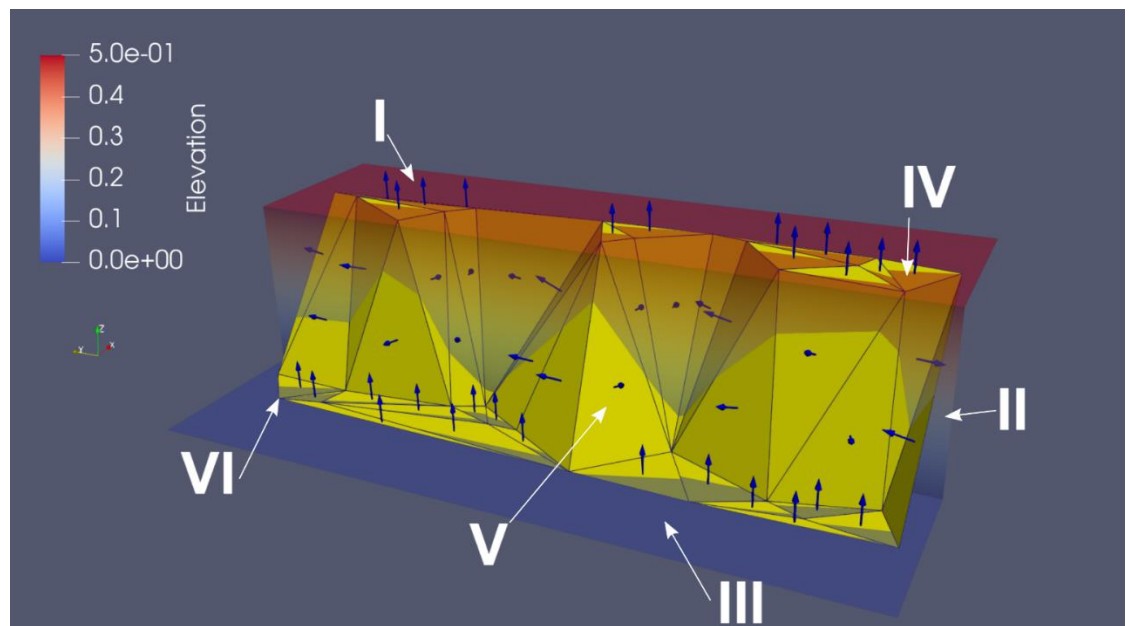


**Figure 1** Presentation of the terminology used in the study: (I) the surface of the horizontal footwall, (II) the fault plane, (III) the surface of the horizontal hanging wall, (IV) a horizontal triangle that is genetically unrelated to the fault, (V) a triangle which is genetically related to the fault (a fault-related triangle), (VI) a vertex of the triangle (a point corresponding to a geological interface identified by a borehole).

## 3.3 Statistical analysis

Treating the normal vectors as 3D directional data makes it possible to calculate the mean of a group of these 3D vectors. It can be achieved by averaging the Cartesian coordinates of the normal vectors. Then, the resultant vector can be converted to dip direction and dip angle pairs (Allmendinger et al., 2011 - Chapter 2.4). We note that in this approach the directional components ($X$ and $Y$ coordinates) are not guaranteed to result in a vector of unit length. Therefore, every vector can contribute





differently to the resultant vector (Fig. 2). For example, 3D vectors corresponding to sub-horizontal triangles, have smaller

values than more inclined triangles at directional components. Therefore, the contribution of sub-horizontal triangles to the

resultant vector will be relatively small compared to more inclined triangles (Fig. 2).

A different approach would be to conduct a statistical analysis of 2D unit vectors $d_1, \ldots d_n$ corresponding to the initially

collected 3D unit normal vectors of triangles $t_1, \ldots, t_n$, where $n$ denotes the number of observations. The mean direction $\overline{\theta}$ of

2D unit vectors $d_1, \ldots d_n$ and their corresponding angles $\theta_1, \ldots \theta_n$ is defined as the direction of the resultant vector $d_1 + \cdots + d_n$

(Mardia and Jupp, 2008). First, the Cartesian coordinates of the centre of the mass (Mardia and Jupp, 2008) are calculated as

follows:

$\overline{C} = \frac{1}{n}\sum_{j=1}^{n} cos\theta_j, \overline{S} = \frac{1}{n}\sum_{j=1}^{n} sin\theta_j$. We note that in our case the $X$ and $Y$ axes are aligned with the North and East directions,

respectively. Therefore, the $\overline{C}$ and $\overline{S}$ values correspond to North and East directions, respectively (a different convention is

adopted in the textbook (Mardia and Jupp, 2008) ). To calculate the mean direction $\overline{\theta}$ , we use the following formula (modified

from Fisher, 1993)

$$
\overline{\theta} = \begin{cases} atan\left(\frac{\overline{S}}{\overline{C}}\right), \overline{S} > 0, \overline{C} > 0 \\ atan\left(\frac{\overline{S}}{\overline{C}}\right) + \pi, \overline{C} < 0 \\ atan\left(\frac{\overline{S}}{\overline{C}}\right) + 2\pi, \overline{S} < 0, \overline{C} > 0 \end{cases} \qquad (Eq.\ 1)
$$

The *resultant length* is the length of the resultant vector sum $d_1 + \cdots + d_n$ and the mean resultant length is defined as the length

of the centre of the mass vector $\overline{R} = \sqrt{\overline{C}^2 + \overline{S}^2}$. We calculated the median direction as well using the circular package (Lund

et al., 2017) and it is any angle $\phi$ such that (Mardia and Jupp, 2008):

i)      half of the data points lie in the arc $[\phi, \phi + \pi]$

ii)     the majority of the data points are nearer to $\phi$ than to $\phi + \pi$.

The sample circular standard deviation is defined as $\sqrt{-2log\ (1-V)} = \sqrt{-2log\overline{R}}$, where $V = 1 - \overline{R}$ denotes the sample

circular variance (Mardia and Jupp, 2008). Using $1 - cos(\theta - \xi)$ as a measure of the distance between angles $\theta$ and $\xi$, it can

be shown that $V$ can be used as a measure of dispersion around the mean dip direction and it is equal to $V = D(\overline{\theta}) =$

$\frac{1}{n}\sum_{j=1}^{n}\{1 - cos\ (\theta_i - \overline{\theta})\}$. We calculate also the sample circular dispersion (Fisher, 1993) defined as $\overline{\delta} = \frac{1-m_2}{(2\overline{R}^2)}$, where $m_2$

denotes the second central trigonometric moment and it is equal to $m_2 = \frac{1}{n}\sum_{j=1}^{n} cos2\ (\theta_i - \overline{\theta})$. We also use the circular

standard error defined as the square root of the sample circular dispersion divided by the number of samples $\overline{\sigma}^2 = \frac{\overline{\delta}}{n}$. Using the

above quantities, non-parametric methods can be used to estimate $100(1 - \alpha)\%$ confidence intervals for $\overline{\theta}$:

$\left(\overline{\theta} - arcsin\left(z_{\frac{1}{2}\alpha}\overline{\sigma}\right), \overline{\theta} + arcsin\left(z_{\frac{1}{2}\alpha}\overline{\sigma}\right)\right)$, where $z_{\frac{1}{2}\alpha}$ is the upper $100\left(\frac{1}{2}\alpha\right)\%$ point of the $N(0,1)$ distribution.




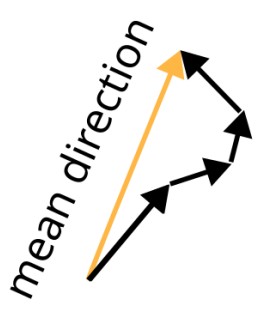 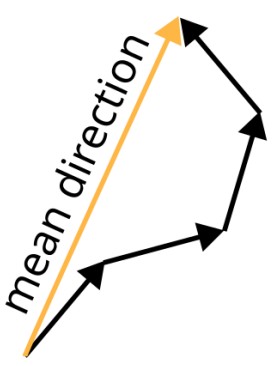


**Figure 2** Illustration of the difference between calculating the mean direction using $X$ and $Y$ components of the 3D and 2D unit vectors. In the first approach, the vectors impact the resulting dip directions differently. Triangles with greater dip angles have a more significant impact, and more horizontally oriented triangles have minor $X$ and $Y$ components making their contribution less important. In the 2D analysis, all vectors are equally important because they all have unit length.

**4. Results**

**4.1 A guide to theoretical results**

This section summarizes the main theoretical findings from the four key propositions, focusing on their implications for fault-related triangles (the workflow of the theoretical part is presented in Fig. 3)

**Proposition 1** shows that the dip direction of fault-related triangles is controlled by the orientation of the fixed edge on the
footwall and the position of the third point on the hanging wall. The dip direction can only be one of two values, differing by 180°, and is perpendicular to the fixed edge. The third point's position (left or right of the edge) determines which direction is observed (Fig. 4).

**Proposition 2** extends this by proving that adding horizontal triangles from the footwall does not alter the mean dip direction of the fault-related triangles. Since the normal vectors of footwall triangles are $[\mathbf{0}, \mathbf{0}, \mathbf{1}]$, they do not influence the dip direction.

**Proposition 3** considers a fixed edge on the footwall and a third point on the hanging wall. It investigates the effect of elevation errors and concludes that the expected dip direction remains unchanged, even when elevation uncertainties are introduced. This demonstrates that fault analysis remains robust despite moderate elevation inaccuracies.



**Proposition 4** generalizes Proposition 2 for cases with elevation uncertainties. Adding triangles with uncertain elevations does not affect the mean dip direction because their $X$ and $Y$ components average to zero.

Together, these propositions show that fault-related triangles reliably indicate dip direction, even in the presence of elevation uncertainties.




**Figure 3** Workflow in the theoretical part of the study **(a)** this panel relates to Proposition 1. In this proposition, an edge $e$ is fixed on the surface of the footwall. Then, the dip directions of triangles depend on whether the third point sampled from the hanging wall lies either to the left or right of $e$. The points on the surfaces of the hanging wall and footwall have constant elevation; **(b)** this panel relates to Proposition 2 and it shows a more general scenario than presented in the (a) panel. Here, for a fixed edge $e$, dip directions of triangles depend on whether

the third point lies either to the left or right of $e$. We do not require from the points to lie on the hanging wall because the $[0, 0, 1]$ vectors from the footwall will not affect the average dip direction of triangles. **(c)** Here, we add uncertainty to the elevation of points (Proposition 3). Then, for a fixed edge $e$, we investigate the average dip direction of triangles with the third point from the hanging wall. **(d)** the last



scenario is the most general scenario (Proposition 4), in which points have uncertain elevations and are no longer required to be sampled from the surface of the hanging wall. In other words, for a fixed edge $e$, the dip directions of triangles depend on whether the third point lies

either to the left or to the right of $e$. We no longer require from points to lie only on the hanging wall. This is because the expected values of $X$ and $Y$ coordinates associated with triangles on the footwall are zero.

## 4.2 Formal results

In the following analysis, we assume that the considered triangles are non-vertical and non-horizontal. The reason is that horizontal triangles give no information about the dip direction, and in the case of vertical triangles, there is no possibility of

deciding which of the two directions corresponds to the direction of the dip.

**Proposition 1**

Let $T$ be a set of non-vertical and non-horizontal triangles genetically related to the fault and $e := \{p_1, p_2\}$ be the fixed edge lying on the footwall. When the difference between the hanging wall and the footwall is constant, then the following facts hold:

A)   There are only two possible dip directions for triangles in $T$: $d_1$ and $d_2$ which have a dip direction difference of 180°

   B)   The two different dip directions $d_1$ and $d_2$ are perpendicular to $e$.

   C)   Moreover, the specific value of the direction $d_1$ or $d_2$ depends on whether the third point $p_3$ located at the hanging wall lies to the left or right of the line containing $e$.

*Proof.* See Appendix B and the corresponding illustration (Fig. 4).





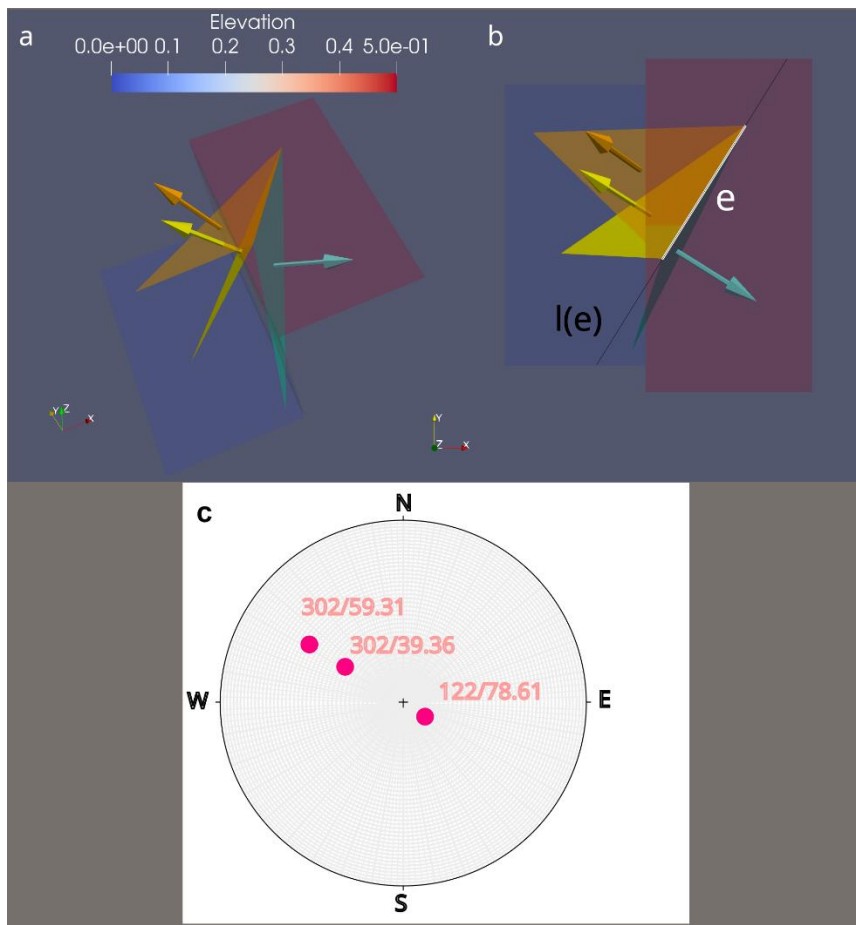

**Figure 4**. Illustration of Proposition 1. There are three triangles that share the same edge on the footwall. Two of the triangles (yellow and orange) have the same dip direction (302). In other words, the projections of the yellow and orange vectors onto the horizontal plane are parallel. We note that $l(e)$ denotes the line containing $e$. Then, the remaining points forming the yellow and orange triangles lie to the left to $l(e)$. The third triangle (pink) has the opposite direction (122) and the third point lies to the right of $l(e)$. On the panel (c), we presented the three orientation measurements from panels (a) and (b) on the spherical projection. The spherical projection was performed using the Stereonet software (Allmendinger et al., 2011; Cardozo and Allmendinger, 2013)



**Observation 1**. For a fixed edge on the footwall and a vertical fault dipping to the West (azimuth 270), the probability that a triangle will dip precisely to the East (azimuth 90) is zero.

*Proof*. For a triangle to dip exactly to the East, one should have an edge aligned with the N-S direction. However, the third point must be to the right of such an edge. However, the hanging wall is to the left of this edge, so no appropriate points can be sampled from the surface of the hanging wall or the footwall.

**Proposition 2.** Extending Proposition 1, we demonstrate that adding horizontal triangles from the footwall does not alter the mean dip direction of fault-related triangles.

*Proof*. The set of triangles on the right side of the edge can be divided into those with the third point on the hanging wall (indices from 1 to $k$) and those on the footwall (vertices from $k + 1$ to $n$).

$$v_1 = [x_1, y_1, z_1] \qquad \text{(Eq. 2)}$$
$$v_2 = [x_2, y_2, z_2]$$
$$\vdots$$
$$v_k = [x_k, y_k, z_k]$$
$$v_{k+1} = [x_{k+1}, y_{k+1}, z_{k+1}] = [0, 0, 1]$$
$$\vdots$$
$$v_n = [x_n, y_n, z_n] = [0, 0, 1]$$

But we know that all vectors from the footwall starting from $v_{k+1}$ to $v_n$ have the form $[0, 0, 1]$ and they contribute nothing to the dip direction.





Next, we can try to adapt and rewrite the Proposition 1 (the third point on the surface of the hanging wall) for uncertain
elevation data. It means that we add normally distributed errors (with an expected value equal to zero) to the elevations. Using
this assumption, the Proposition 1 can be rewritten in the following way.

**Proposition 3.**

When introducing elevation uncertainties, we find that the expected dip direction remains consistent with the error-free case.
The third point is required to be from the surface of the hanging wall.

*Proof.* See Appendix C.

**Proposition 4.**

This proposition generalizes Proposition 3, showing that adding footwall triangles with uncertain elevations still does not affect
the mean dip direction. The $X$ and $Y$ components of the footwall triangles' normal vectors average to zero, further confirming
the robustness of the method under elevation uncertainties.

*Proof.* As in the deterministic example (Proposition 2), the set of triangles on the right side of the edge can be divided into
those with the third point on the hanging wall (indices from 1 to $k$) and those with the third point on the footwall (vertices
from $k + 1$ to $n$). To complete the proof, we need to show that the expected values of the $X$ and $Y$ coordinates of the normal
vectors representing triangles with third points lying on the surface of the footwall (from $v_{k+1}$ to $v_n$) are zero. We present the
desired proofs in Appendix D.

$$v_1 = [x_1, y_1, z_1] \qquad \text{(Eq. 3)}$$
$$v_2 = [x_2, y_2, z_2]$$
$$\vdots$$
$$v_k = [x_k, y_k, z_k]$$
$$v_{k+1} = [x_{k+1}, y_{k+1}, z_{k+1}]$$
$$\vdots$$
$$v_n = [x_n, y_n, z_n]$$
This implies that the sums of the $X$ and $Y$ coordinates from $v_{k+1}$ to $v_n$ are zero: $\sum_{k+1}^n x_i = \sum_{k+1}^n y_i = 0$. Therefore, as in the
deterministic case, adding these sums will not change the first two coordinates of the expected normal vector. Appendix D
provides further discussion on when the mean vector from $v_{k+1}$ to $v_n$ is parallel to $[0, 0, 1]$.






### 4.3 Experimental results

Our first case study involves irregularly scattered points, commonly observed in borehole data sets, with a fault dipping to the West (Fig. 5). To better understand the impact of preferred edge orientation and the imbalance between points on the hanging wall and footwall, we also examined regularly scattered data with a fault trending obliquely to the main grid axes (Fig. 6). In these case studies, the true (ground truth) orientations of the faults were 270 and 340.43 degrees, respectively, which can be compared against the calculated statistics (Table 1).

Standard statistical methods, including the estimation of confidence intervals, demonstrated promising utility in inferring the true dip direction. For example, the minor deviations of 1-2 degrees between error-free and with-error scenarios suggest that the model is robust in the presence of elevation errors, reinforcing the conclusions from Proposition 4. These minor deviations indicate that the method provides reliable dip direction estimates even under uncertain elevation conditions.

However, limitations remain. The confidence intervals did not contain the true dip direction, with a mismatch ranging from 2 to 5 degrees. It is unclear whether this mismatch is due to the dataset limitations or the elevation errors' specific characteristics. Although both the directional dispersion and circular standard error increased slightly in the case studies with uncertain elevations (Table 1), the increase was not significant, and the circular standard error remained low, resulting in very narrow confidence intervals.



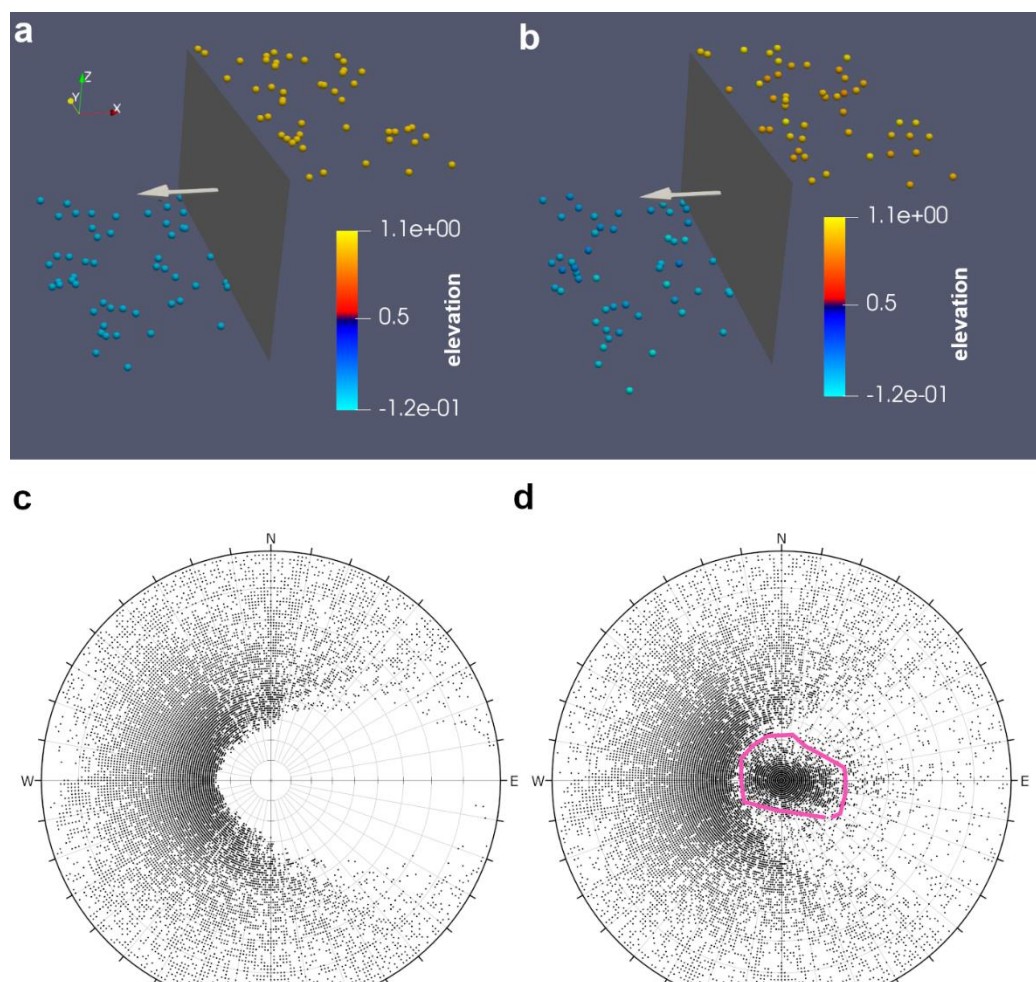

**Figure 5** A geological horizon displaced by a vertical fault; its normal vector points to the West (azimuth 270) **(a)** a case study with elevation data without errors; **(b)** a case study with points having elevation data with errors (mean=0, standard deviation=0.05), **(c)** a spherical projection corresponding to data presented on panel (a), n=122304, **(d)** a spherical projection corresponding to data (n=161700) presented on panel (b). The small cloud at the centre of the plot (pink polygon) corresponds to almost flat triangles lying entirely on the same side of the fault. The spherical projection was conducted using Dips software (Rocscience, 2017). We used pole vectors, upper hemisphere and equal angle projection.





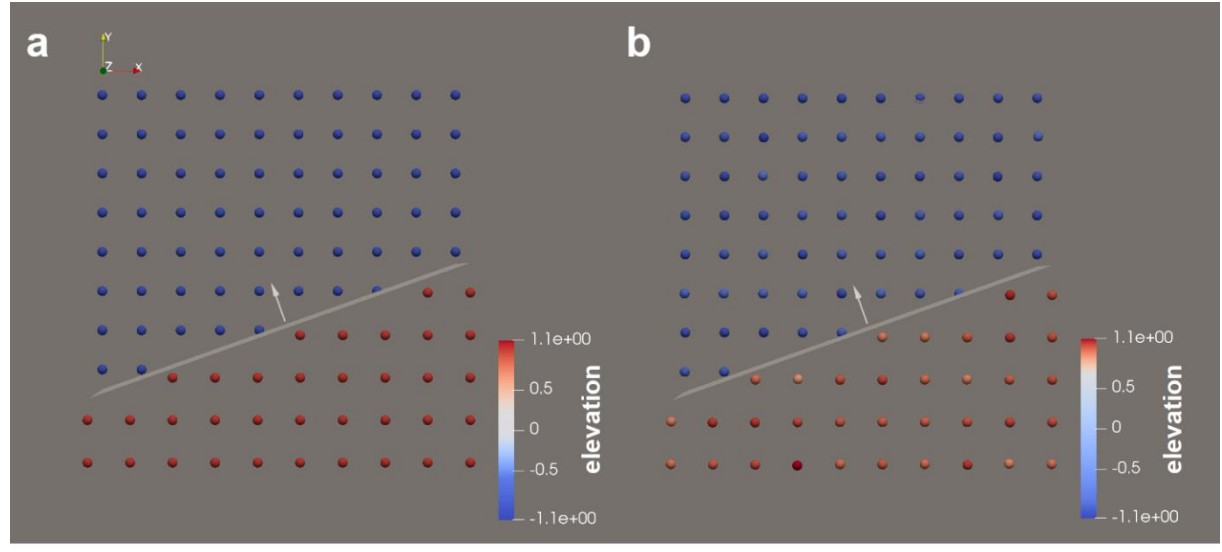

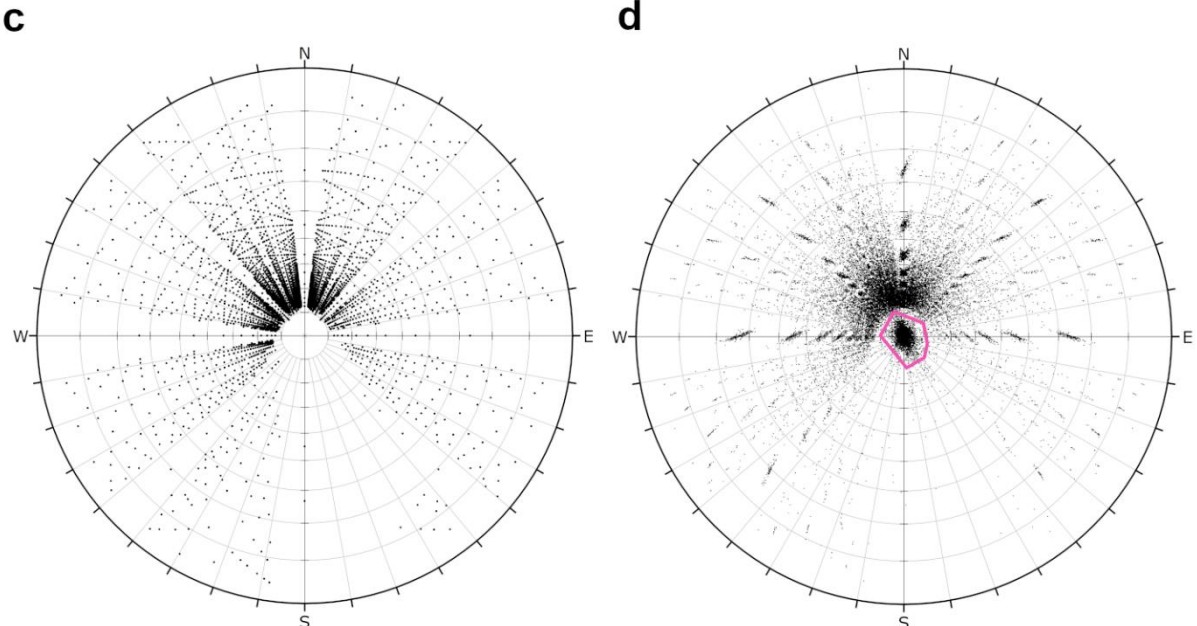

**Figure 6** Orientation measurements using a combinatorial algorithm for a regular grid of points and a vertical fault trending obliquely to the main axes of the grid. The vertical fault has a normal vector with azimuth 340.43. **(a)** input points without elevation errors, **(b)** input points with elevation errors (mean=0, standard deviation=0.05), **(c)** a spherical projection for the orientation measurements (n=109111) without elevation errors corresponding to panel (a), **(d)** a spherical projection for the orientation measurements (n=157252) with elevation errors corresponding to panel (b). The small cloud at the centre of the plot (pink polygon) corresponds to almost flat triangles lying entirely on the same side of the fault. The spherical projection was conducted using Dips software (Rocscience, 2017). We used pole vectors, upper hemisphere and equal angle projection.






**Table 1** Statistical results for the case studies presented in Figs 5 and 6.

|  | **Without error (Fig. 5a,c)** | **With error (Fig. 5b,d)** | **Without error (Fig. 6a,c)** | **With error (Fig. 6b,d)** |
|---|---|---|---|---|
| Number of observations | 122304 | 161698 | 109111 | 157175 |
| Mean direction (3D) | 276.35 | 275.02 | 343.16 | 343.39 |
| Mean direction (2D) | 275.91 | 273.28 | 343.84 | 345.08 |
| Median direction (2D) | 276.16 | 274.04 | 345.96 | 345.81 |
| Resultant length (2D) | 84808.32 | 77634.55 | 76177.25 | 78671.65 |
| Mean resultant length (2D) | 0.69 | 0.48 | 0.70 | 0.50 |
| Sample circular variance (2D) | 0.31 | 0.52 | 0.30 | 0.50 |
| Circular standard deviation (2D) | 0.86 | 1.21 | 0.85 | 1.18 |
| Sample circular dispersion (2D) | 0.93 | 1.98 | 0.92 | 1.82 |
| Circular standard error (2D) | 0.0028 | 0.0035 | 0.0029 | 0.00339 |
| 95% confidence intervals (2D) | [275.60-276.22] | [272.89, 273.68] | [343.51, 344.16] | [344.70, 345.47] |





## 5. Discussion

### 5.1. New insights from geometric and data analysis

In planar structures such as triangles, the strike is defined by the intersection of the triangle and a horizontal plane. Therefore, a triangle with a flat-lying edge can be interpreted as having a strike parallel to this edge. The determination of the dip direction for such triangles requires solving a computational geometry problem: determining whether the third point lies to the left or right of this flat-lying edge.

Our study shows that flat-lying edges parallel to the fault strike are privileged due to the greater number of points lying on one side of the edge. As a result, these directions carry more weight in statistical calculations and are more likely to represent the true dip direction. Identifying these privileged dip directions is essential for accurate fault orientation predictions, as the concentration of observations around the true dip direction indicates a reliable methodology.

At this stage, it may be beneficial to complement standard statistical approaches with qualitative observations based on the
distribution of dip directions. For example, in Figs. 5c,d and 6c,d, the directions opposite to the true dip direction (90 and 160.43 degrees) are sparsely represented, which may help constrain the interpretation by highlighting directions that are unlikely to represent the true dip. This pattern aligns with the formal results from Proposition 1, which indicate that for a fixed edge parallel to the fault strike on the upper wall (hanging wall), no dip directions will point precisely toward the upper wall (Observation 1). This observation provides additional confidence in the reliability of the predicted dip directions.


### 5.2 Comparison with similar approaches

In most studies related to geological model uncertainty, multiple faults are considered to estimate the uncertainty of the fault network or geometry (Cherpeau and Caumon, 2015; Lecour et al., 2001). The variability of faults is typically expressed through their parametrization, which often includes the strike and dip of the fault (Cherpeau et al., 2012; Aydin and Caers, 2017;
Goodwin et al., 2022). In contrast, in this study, we set the orientation of the fault constant to provide ground truth data, enabling us to investigate the mathematical relationships between points and dip directions of triangles genetically related to the fault.

By keeping the fault orientation constant, we could isolate the effects of data uncertainties and mathematically investigate the relationship between points and dip directions. This controlled environment provides a more precise assessment of how data
uncertainties, specifically elevation errors, influence dip direction calculations without the added complexity of variable fault parameters.

To investigate the impact of data uncertainties on the calculated fault orientation, we added errors to the elevation values (Z coordinate), following a normal probability distribution with a mean equal to the measured elevation. In our case, the



geographical coordinates ($X$ and $Y$) are assumed to be known. Introducing uncertainties in the geographical coordinates, where

locations follow a normal probability distribution (Allmendinger et al., 2011), would present a significant challenge; when

coordinates are uncertain, it becomes difficult to define whether a point lies to the left or right of a line, as the vertices of that

line are also uncertain.





### 5.3 Importance of the results in modelling spatial data

The results of this study can be applied in GIS-based directional and statistical analyses of topographic vector data, such as Triangulated Irregular Network (TIN) objects frequently used in GIS software (e.g., ESRI ArcGIS). In particular, our findings support explaining singular statistical effects in azimuthal analyses of regular topographic data. To support this claim, we note the following:

1. **TIN models** are used in the analysis of bathymetric GEBCO (Gridded Bathymetry Data, 2024) data (Fig. 7a) (Włodarczyk-
Sielicka et al., 2022; Idzikowska et al., 2024)

2. **GEBCO data points** are often stored in quasi-regular point grids. Although these points are regularly spaced in geographic coordinates, when projected into a Cartesian system, the individual points are no longer regularly aligned. This leads to the formation of quasi-parallel edges in the TIN model (Fig. 7b). This quasi-regular alignment of points often results in triangles with nearly parallel edges, creating preferred directions for dip calculations.

3. **Limited precision of the elevation measurements** can lead to the rounding of elevation values to integers. If points within a specified neighborhood are recorded as having only two distinct integer values, then it results in a constant elevation difference between two sets of points. This rounding can lead to flat edges (edges with two identical elevation values) or flat triangles (triangles with three identical elevation values), as marked in Fig. 7b.

For the specific combination of models, spatial distribution of points, and limited precision described above, the conditions
from **Proposition 1** may apply, explaining the concentration of dip direction values in azimuthal histograms (Fig. 7c). Only eight narrow groups of azimuth values are present in the entire dataset due to the eight preferred triangle edge directions in the TIN. Two groups (130° and 313°) are significantly underrepresented, likely due to the lack of flat edges striking NE-SW. This feature is purely based on the data representation. Users must take this into account during analysis to avoid skewed interpretations of fault orientations.




**Figure 7** Triangulated network of example bathymetric GEBCO data stored in a quasi-regular grid **(a)** a repeating valley-ridge pattern with a marked rectangle analyzed on the panel (b); it is located at a curved axis of a negative landform as testified by opposing dip directions near the axis**; (b)** a zoom at the rectangle marked on the panel (a); here we can see that triangles with the same elevations of nodes yield a flat triangle; triangles with a flat lying edge but a third vertex with a different elevation than the other two points have dip direction perpendicular

to the flat edge; the interiors of triangles are filled with the color corresponding to the value of azimuth from panel (a); **(c)** a histogram of circular data related to the TIN model. The histogram shows narrow azimuthal groups with approximately 4° spacing corresponding to the edges of a regular grid. The length and color of each segment represent the amount of triangles in the corresponding group.



**6 Conclusion**

This study aimed to bridge computational geometry and structural geology to explore the behavior of triangles related to a horizon displaced by a vertical fault. We conducted analyses under both idealized (two elevations) and uncertain conditions (added elevation error). A key challenge was adapting concepts from computational geometry, such as determining whether a point lies to the left or right of a line, to a geological context.

Key findings from the study include:

- This study shows that for TIN-based modeling, determining the dip direction of a triangle with a flat-lying edge requires solving a computational geometry problem: determining whether the third point lies to the left or right of the edge.

- The problem of assessing fault orientation can be approached as an optimization task, where the fault orientation is
estimated by identifying the edge with the maximum number of points on one side, leading to the maximum number of triangles with the same dip direction.

- Our formal analysis shows that introducing measurement errors does not affect the expected dip direction of samples, which remains identical to the error-free case. Moreover, the statistical results and orientation distributions remain robust across different fault and study area orientations, suggesting practical applications.

- The importance of these findings is particularly relevant for directional analyses of imprecise topographic data, such as azimuth maps of bathymetric point datasets distributed in regular grids. The concentration of azimuths around the N-S, W-E, NE-SE, and NE-SW directions corresponds directly to the triangulation's edges in the regular grid.






## Appendices

**Appendix A.**

To generate all possible triangles from a given set of boreholes, we used an algorithm to generate all $k$-element ($k = 3$) subset from an $n$-element set $X$ ($k < n$) in a lexicographic order (Lipski, 2004). To explain how the algorithm works, we first note that every $k$-element subset can be uniquely represented by an increasing sequence of length $k$ of elements from $X$. For example, the subset $\{3, 5, 1\}$ can be represented as a sequence $(1, 3, 5)$. The first step in the algorithm involves writing the

first $k$ digits from $X$. For example, if $k = 4$, the first sequence would be $(1, 2, 3, 4)$. Then, the sequence succeeding $(a_1, \dots, a_k)$ is

$$(b_1, \dots, b_k) = (a_1, \dots, a_{p-1}, a_p + 1, a_p + 2, \dots, a_p + k - p + 1), \quad \text{(Eq. 4) where}$$

$$p = max\{i: a_i < n - k + i\}. \quad \text{(Eq. 5)}$$

Likewise, the sequence which succeeds $(b_1, \dots, b_k)$ is

$$(c_1, \dots, c_k) = (b_1, \dots, b_{p'-1}, b_{p'} + 1, b_{p'} + 2, \dots, b_{p'} + k - p' + 1), \quad \text{(Eq. 6)}$$

where

$p' = p - 1$ when $b_k = n$ and

$p' = k$ when $b_k < n$. (Eq. 7)

During the procedure, we assume that the sequences $(a_1, \dots, a_k)$ and $(b_1, \dots, b_k)$ are different from $(n - k + 1, \dots, n)$, the last

sequence in our order. Here, $p$ and $p'$ can be conceptualized as indices where updates of digits starting from the largest $(k)$ index terminate. If $p$ or $p'$ are determined, then $a_p$ or $b_{p'}$ can easily be found, allowing to use it in the update procedure (Eq. 4 and Eq. 6). We note that the number of $k$-element subsets from an $n$-element set can be determined using the binomial coefficient. For example, if $n = 6$ and $k = 4$, then

$$\binom{n}{k} = \frac{n!}{k!(n-k)!} = \frac{4! * 5 * 6}{4! * 2!} = \frac{30}{2} = 15. \quad \text{(Eq. 8)}$$


**Appendix B.** A formal proof of Proposition 1.

**Definitions**

First, we remind the definitions given in the article (Michalak et al., 2021):

**Property 1**. (property of the models). A triangle is not horizontal if and only if not all of the three vertices lie on the same side

of the fault.

Equivalently, Property 1 can be expressed as follows: A triangle is not horizontal if and only if there exists a pair of vertices that do not lie on the same side of the fault.





**Definition 1**. (triangles genetically related to a fault in the model). A triangle is referred to as genetically related to fault if it is not horizontal.

**Proposition 1**

*Proof.* We note that, intuitively, the result from parts A and B is in line with the standard concept of a strike of a geological planar feature defined as the intersection of this planar feature and a horizontal plane (Fossen, 2006, see also Fig. 4 as an illustration of the proposition).

To formally prove the proposition, in particular part C, we will first refer to the following orientation test (De Berg et al.,

405 2008):

**Fact 1** (orientation test). (De Berg et al., 2008)

The sign of the determinant

$$|D| = det \begin{matrix} 1 & t_1 & t_2 \\ 1 & u_1 & u_2 \\ 1 & s_1 & s_2 \end{matrix} \qquad (Eq. 9)$$

determines whether $s$ lies left or right of the line $tu$.

*Proof of Proposition 1.*

  *Part A of Proposition 1.*

Let $p_1 = (x_1, y_1, z_1)$, $p_2 = (x_2, y_2, z_2)$, be points forming an edge. For simplicity, let's assume that $p_1$ and $p_2$ are located on the surface of the footwall. The third point $p_3 = (x_3, y_3, z_3)$ can be anywhere on the surface of the hanging wall. Therefore, we will consider a set of many triangles. Because the walls are horizontal and the difference between walls is constant, we can

write $p_1 = (x_1, y_1, a)$, $p_2 = (x_2, y_2, a)$, and $p_3 = (x_3, y_3, b)$, where $a \neq b$. Let $k := a - b$ be the positive constant being the elevation difference between walls.

The two vectors spanning the triangle's plane are as follows:

$$v_1 = [x_2 - x_1, y_2 - y_1, 0], \qquad (Eq. 10)$$
$$v_2 = [x_3 - x_1, y_3 - y_1, -k], \qquad (Eq. 11)$$

Using the cross product, the coordinates of the normal vector of the triangle defined by the above vectors can be calculated via the mnemonic rule:

$$det \begin{matrix} x_2 - x_1 & x_3 - x_1 & n_1[1] \\ y_2 - y_1 & y_3 - y_1 & n_1[2] \\ 0 & -k & n_1[3] \end{matrix} = (x_2 - x_1) * (y_3 - y_1) * n_1[3] + (y_2 - y_1) * -k * n_1[1] + 0 * (x_3 - x_1) * n_1[2] -$$

$$n_1[1] * (y_3 - y_1) * 0 - n_1[2] * -k * (x_2 - x_1) - n_1[3] * (x_3 - x_1) * (y_2 - y_1) \qquad (Eq. 12)$$

In summary:

$n_1[1] = (y_2 - y_1) * -k - (y_3 - y_1) * 0 = (y_2 - y_1) * -k = -k * (y_2 - y_1)$ (Eq. 13)

$n_1[2] = 0 * (x_3 - x_1) - (-k) * (x_2 - x_1) = k * (x_2 - x_1)$ (Eq. 14)

$n_1[3] = (x_2 - x_1) * (y_3 - y_1) - (x_3 - x_1)(y_2 - y_1) = x_2 y_3 - x_2 y_1 - x_1 y_3 + x_1 y_1 - x_3 y_2 + x_3 y_1 + x_1 y_2 = x_2 y_3 +$

$x_3 y_1 + x_1 y_2 - x_2 y_1 - x_1 y_3 - x_3 y_2$ (Eq. 15)





As of now, we have two notes:

430  - Every non-vertical triangle has two normal vectors: one is directed downwards and the other is directed upwards. We are only interested in normal vectors directed upwards to avoid duplicate representations and ensure consistent representation of observations.

   - Note that only the edge on the footwall is fixed and not the third point whose coordinates affect the sign of the third coordinate of the normal vector (Eq. 15). Therefore, it can be concluded that we don't investigate a particular

435    normal vector but a set of many normal vectors. Moreover, given only the position of the fixed edge, the normal vector's third coordinate is unknown, and three cases must be considered: when it is positive, negative, or zero. We will now consider these three cases.

  I. If $n_1[3] > 0$, then the coordinates of the normal vector look as above, which means that the vector is directed upwards.

440  II. If $n_1[3] < 0$, then it means that the normal vector is directed downwards, and the coordinates must be multiplied by minus 1 to adhere to the above rule that there is only one representation of normal vectors.

  III. If $n_1[3] = 0$, then the normal vector is directed horizontally (orthogonal to the vectors $v_1 = [x_2 - x_1, y_2 - y_1]$ and $v_2 = [x_3 - x_1, y_3 - y_1]$ and the corresponding triangle is vertical, contrary to the initial assumption of considering only non-vertical triangles. Therefore, we no longer consider this scenario.

445 Now, according to (II), we multiply coordinates from Eqs. 13-15 by minus 1.

$n_2[1] = k * (y_2 - y_1)$ (Eq. 16)

$n_2[2] = -k * (x_2 - x_1)$ (Eq. 17)

$n_2[3] = -(x_2 y_3 + x_3 y_1 + x_1 y_2 - x_2 y_1 - x_1 y_3 - -x_3 y_2)$   (Eq. 18)

We observe that the coordinates of normal vectors (Eqs 13-15 and Eqs 16-18) are not the same which means that we obtained

450 two distinct normal vectors directed upwards $n_1$ and $n_2$ and which have a dip direction difference of $180°$. This is because $n_1[1] = -n_2[1]$ and $n_1[2] = -n_2[2]$. The vectors are directed upwards because the third coordinate is positive. So, part A of the Proposition 1 is proven.

  *Part B of Proposition 1.*

As we already know from A), there are only two possible dip directions of the infinite set of triangles. These dip directions

455 have a directional difference of $180°$. Therefore, to prove that they are perpendicular to the edge $p_1 p_2 = [x_2 - x_1, y_2 - y_1]$, it is enough to prove it for one vector: $n_1$ or $n_2$ projected on the horizontal plane (the projection of the normal vector onto the horizontal plane doesn't change its direction). Using the dot product (·), we can show it for $\breve{n}_1$ , where $\breve{n}_1$ denotes the projection of the first normal vector onto the horizontal plane:

$p_1 p_2 \cdot \breve{n}_1 = (x_2 - x_1) * -k * (y_2 - y_1) + (y_2 - y_1) * k * (x_2 - x_1) = -k(x_2 - x_1)(y_2 - y_1) + k(x_2 - x_1)(y_2 - y_1) = 0$

460        (Eq. 19)





*Part C of Proposition 1.*

We have a fixed edge on the footwall, and we can consider two cases: when the point lies to the left or to the right of the line
465    containing the edge.

Using Fact 1 (orientation test), we can determine whether $p_3 = (x_3, y_3)$ lies to the left, to the right, or on the edge $p_1 p_2$:

$$|D| = det\begin{vmatrix} 1 & x_1 & y_1 \\ 1 & x_2 & y_2 \\ 1 & x_3 & y_3 \end{vmatrix} = x_2 y_3 + x_3 y_1 + x_1 y_2 - y_1 x_2 - y_2 x_3 - y_3 x_1 \qquad \text{(Eq. 20)}$$

We note that the value can be positive, negative, or zero depending on whether the point $p_3$ lies left, right, or on the edge $p_1 p_2$.
Recall that in the part A this value was also the third coordinate of the first non-horizontal normal vector $n_1[3]$ and the additive
470    inverse $-(x_2 y_3 + x_3 y_1 + x_1 y_2 - x_2 y_1 - x_1 y_3 - x_3 y_2)$ was the third coordinate of the second normal vector. Therefore, the
signs of the expressions $sgn(x_2 y_3 + x_3 y_1 + x_1 y_2 - y_1 x_2 - y_2 x_3 - y_3 x_1) \neq sgn\big(-(x_2 y_3 + x_3 y_1 + x_1 y_2 - y_1 x_2 - y_2 x_3 -$
$y_3 x_1)\big)$ simultaneously determine the position of the point $p_3$ relative to the edge $p_1 p_2$ and the choice of one of two possible
normal vectors. In the case of |**D**|=0 the points $p_1, p_2$ (footwall) and $p_3$ (hanging wall) are collinear in 2D space but not in 3D
space and the corresponding triangle is vertical, which is beyond the scope of our study.





**Appendix C.**

**Proposition 3.** *Analysis with elevation uncertainties restricted to the hanging wall regarding the free vertices.*

Similar calculations can be conducted for data with fixed geographical position but uncertain elevations. To achieve this, the uncertain elevations can be represented as sums of the measured constant elevations and a random variable $\varepsilon$ with the normal distribution $N(0, \sigma^2)$.

Therefore, the uncertain elevations of *n* points are independent random variables $\varepsilon_1, \ldots, \varepsilon_n$ and their expected values are equal to zero, i.e., $E[\varepsilon_1] = \cdots = E[\varepsilon_n] = 0$. Here we consider $x_1, y_1, z_1, x_2, y_2, z_2$ as fixed constants.

From a practical viewpoint, we first create the set of points from the uniform distribution displaced by a vertical fault and then add error to elevation data.

From then on, we have points $p_1 = (x_1, y_1, z_1 + \varepsilon_1)$, $p_2 = (x_2, y_2, z_1 + \varepsilon_2)$ forming an edge on the footwall. The third point $p_3 = (x_3, y_3, z_2 + \varepsilon_3)$ can lie anywhere on the hanging wall. For now, we consider the point $p_3$ to be fixed, but ultimately, it will traverse the points on the hanging wall to compute their average.

The random vectors spanning the plane of a random triangle are calculated as follows:

$$\boldsymbol{v_1} = [\boldsymbol{x_2} - \boldsymbol{x_1}, \boldsymbol{y_2} - \boldsymbol{y_1}, \boldsymbol{\varepsilon_2} - \boldsymbol{\varepsilon_1}], \text{ and let } \alpha := \varepsilon_2 - \varepsilon_1. \text{ Then } \boldsymbol{v_1} = [\boldsymbol{x_2} - \boldsymbol{x_1}, \boldsymbol{y_2} - \boldsymbol{y_1}, \boldsymbol{\alpha}], \quad \text{(Eq. 21)}$$

$$\boldsymbol{v_2} = [\boldsymbol{x_3} - \boldsymbol{x_1}, \boldsymbol{y_3} - \boldsymbol{y_1}, \boldsymbol{z_2} + \boldsymbol{\varepsilon_3} - \boldsymbol{z_1} - \boldsymbol{\varepsilon_1}], \text{ and let } k := -(z_2 - z_1) \text{ with } \beta := \varepsilon_3 - \varepsilon_1.$$

$$\text{Then } \boldsymbol{v_2} = [\boldsymbol{x_3} - \boldsymbol{x_1}, \boldsymbol{y_3} - \boldsymbol{y_1}, -\boldsymbol{k} + \boldsymbol{\beta}], \quad \text{(Eq. 22)}$$

The normal vector can be calculated using the mnemonic rule for cross product:

$$det\begin{matrix} x_2 - x_1 & x_3 - x_1 & n_1[1] \\ y_2 - y_1 & y_3 - y_1 & n_1[2] \\ \alpha & -k + \beta & n_1[3] \end{matrix} = (x_2 - x_1) * (y_3 - y_1) * n_1[3] + (y_2 - y_1) * (-k + \beta) * n_1[1] + \alpha * (x_3 - x_1) * n_1[2] -$$

$$n_1[1] * (y_3 - y_1) * \alpha - (x_2 - x_1) * (-k + \beta) * n_1[2] - (y_2 - y_1) * (x_3 - x_1) * n_1[3] \quad \text{(Eq. 23)}$$

In summary,

$$n_1[1] = (-k + \beta) * (y_2 - y_1) - \alpha * (y_3 - y_1) \quad \text{(Eq. 24)}$$

$$n_1[2] = (k - \beta) * (x_2 - x_1) + \alpha * (x_3 - x_1) \quad \text{(Eq. 25)}$$

$$n_1[3] = (x_2 - x_1) * (y_3 - y_1) - (x_3 - x_1)(y_2 - y_1) \quad \text{(Eq. 26)}$$

As in the deterministic case, we note that the third coordinate can be positive or negative. If it is positive, then the coordinates stay as they are. Otherwise, all coordinates must be multiplied by minus 1. Then, the coordinates are as follows:

$$n_2[1] = (k - \beta) * (y_2 - y_1) + \alpha * (y_3 - y_1) \quad \text{(Eq. 27)}$$

$$n_2[2] = (-k + \beta) * (x_2 - x_1) - \alpha * (x_3 - x_1) \quad \text{(Eq. 28)}$$

$$n_2[3] = -(x_2 - x_1) * (y_3 - y_1) + (x_3 - x_1)(y_2 - y_1) \quad \text{(Eq. 29)}$$

For both cases, we can now calculate the expectations of the first two coordinates, given that the above expressions contain random variables.





The first case (positive Z value):

$$E[n_1[1]] = \qquad\qquad \text{(Eq. 30)}$$
$$E[(-k + \beta) * (y_2 - y_1) - \alpha * (y_3 - y_1)] =$$
$$(-\text{k} + E[\beta]) * (y_2 - y_1) - E[\alpha] * (y_3 - y_1) =$$
$$(-\text{k} + (E[\varepsilon_3] - E[\varepsilon_1])) * (y_2 - y_1) - (E[\varepsilon_2] - E[\varepsilon_1]) * (y_3 - y_1) =$$
$$-k * (y_2 - y_1).$$

Since $E[n_1[1]]$ does not depend on $y_3$, averaging over all hanging wall points does not change the expected value, which will remain equal to $-k * (y_2 - y_1)$.

$$E[n_1[2]] = \qquad\qquad \text{(Eq. 31)}$$
$$E[(k - \beta) * (x_2 - x_1) + \alpha * (x_3 - x_1)] =$$
$$(k - (E[\varepsilon_3] - E[\varepsilon_1])) * (x_2 - x_1) + (E[\varepsilon_2] - E[\varepsilon_1]) * (x_3 - x_1) =$$
$$k * (x_2 - x_1).$$

Since $E[n_1[2]]$ does not depend on $x_3$, averaging over all hanging wall points does not change the expected value, which will remain equal to $k * (x_2 - x_1)$.

The second case (negative Z value):

$$E[n_2[1]] = \qquad\qquad \text{(Eq. 32)}$$
$$E[(k - \beta) * (y_2 - y_1) + \alpha * (y_3 - y_1)] =$$
$$k * (y_2 - y_1).$$

As previously, $E[n_2[1]]$ does not depend on $y_3$, averaging over all hanging wall points does not change the expected value, which will remain equal to $k * (y_2 - y_1)$.

$$E[n_2[2]] = \qquad\qquad \text{(Eq. 33)}$$
$$E[(-k + \beta) * (x_2 - x_1) - \alpha * (x_3 - x_1)] =$$
$$-k * (x_2 - x_1)$$

Again, $E[n_2[2]]$ does not depend on $x_3$, averaging over all hanging wall points does not change the expected value, which will remain equal to $-k * (x_2 - x_1)$.





**Appendix D.**

**Proposition 4.** *Analysis with elevation uncertainties with free points on the surfaces of the hanging wall or footwall.*

Here, we continue the analysis from the Sect. 4.2 by considering only triangles with third points lying on the surface of the footwall (from $v_{k+1}$ to $v_n$). As we have $n$-$k$ triangles, we need $n$-$k$+2 points on the footwall. Denote them by $p_i = (x_i, y_i, z_i + \varepsilon_i)$, where $i = 1,2, \dots, n - k + 2$.

We fix the edge $e$ on the surface of the footwall defined by two points: $p_1 = (x_1, y_1, z_1 + \varepsilon_1)$ and $p_2 = (x_2, y_2, z_1 + \varepsilon_2)$. Because the edge $e$ is fixed, the associated coordinates $x_1, y_1, z_1, x_2, y_2, z_2$ are considered fixed constants. The third point $p_3$ is sampled from the surface of the footwall. As previously, we fix the point $p_3$ for a moment, but ultimately, it will traverse the points on the hanging wall to compute their average.

The third point $p_3 = (x_3, y_3, z_1 + \varepsilon_3)$ is sampled from the surface of the footwall. We calculate the spanning vectors:

$v_1 = [x_2 - x_1, y_2 - y_1, \alpha]$ and $v_2 = [x_3 - x_1, y_3 - y_1, \beta]$, with $\alpha \coloneqq \varepsilon_2 - \varepsilon_1$ and $\beta \coloneqq \varepsilon_3 - \varepsilon_1$.

The normal vector can be calculated using the mnemonic rule for cross product:

$$det \begin{matrix} x_2 - x_1 & x_3 - x_1 & n_1[1] \\ y_2 - y_1 & y_3 - y_1 & n_1[2] \\ \alpha & \beta & n_1[3] \end{matrix} = (x_2 - x_1) * (y_3 - y_1) * n_1[3] + (y_2 - y_1) * (\beta) * n_1[1] + \alpha * (x_3 - x_1) * n_1[2] -$$

$$n_1[1] * (y_3 - y_1) * \alpha - (x_2 - x_1) * (\beta) * n_1[2] - (y_2 - y_1) * (x_3 - x_1) * n_1[3] \qquad \text{(Eq. 34)}$$

In summary,

$$n_1[1] = \beta * (y_2 - y_1) - \alpha * (y_3 - y_1) \qquad \text{(Eq. 35)}$$

$$n_1[2] = -\beta * (x_2 - x_1) + \alpha * (x_3 - x_1) \qquad \text{(Eq. 36)}$$

$$n_1[3] = (x_2 - x_1) * (y_3 - y_1) - (x_3 - x_1) * (y_2 - y_1) \qquad \text{(Eq. 37)}$$

As previously, we start the calculations of excepted values of $E[n_1[1]], E[n_1[1]], E[n_1[3]]$, and then the averages over all footwall points.

$$E[n_1[1]] = \qquad \text{(Eq. 38)}$$
$$E[\beta * (y_2 - y_1) - \alpha * (y_3 - y_1)] =$$
$$E[\beta](y_2 - y_1) - E[\alpha](y_3 - y_1) = 0$$

Since E[$n_1$[1]]=0, the average over all footwall points does not change and will remain equal to 0.

$$E[n_1[2]] = \qquad \text{(Eq. 39)}$$
$$E[-\beta * (x_2 - x_1) + \alpha * (x_3 - x_1)]=$$
$$-E[\beta](x_2 - x_1) + E[\alpha](E[x_3 - x_1]) = 0$$

Since E[$n_1$[2]]=0, the average over all footwall points does not change and will remain equal to 0.



$$E[n_1[3]] = \qquad \text{(Eq. 40)}$$
$$(x_2 - x_1) * (y_3 - y_1) - (x_3 - x_1) * (y_2 - y_1).$$

Now, taking the average over all footwall points, we get:

$$(x_2 - x_1) * \frac{(y_3-y_1)+\cdots+(y_{n-k+2}-y_1)}{n-k} - \frac{(x_3-x_1)+\cdots+(x_{n-k+2}-x_1)}{n-k} * (y_2 - y_1) \quad \text{(Eq. 41)}$$

Observe that $E[n_1[3]] = 0$ if and only if

$$(x_2 - x_1) * [(y_3 - y_1) + \cdots + (y_{n-k+2} - y_1)] = [(x_3 - x_1) + \cdots + (x_{n-k+2} - x_1)] * (y_2 - y_1), \qquad \text{(Eq. 42)}$$

i.e., if and only either $\boldsymbol{w} = [(\boldsymbol{x_3} - \boldsymbol{x_1}) + \cdots + (\boldsymbol{x_{n-k+2}} - \boldsymbol{x_1}), (\boldsymbol{y_3} - \boldsymbol{y_1}) + \cdots + (\boldsymbol{y_{n-k+2}} - \boldsymbol{y_1})]$ is the zero vector or it is non-zero and parallel to the vector $[\boldsymbol{x_2} - \boldsymbol{x_1}, \boldsymbol{y_2} - \boldsymbol{y_1}]$, which happens very seldom.

One of the trivial cases for which $E[n_1[3]] \neq 0$ is when

$$\frac{(x_3-x_1)+\cdots+(x_{n-k+2}-x_1)}{x_2-x_1} \neq \frac{(y_3-y_1)+\cdots+(y_{n-k+2}-y_1)}{y_2 - y_1}, \qquad \text{(Eq. 43)}$$

assuming $x_2 \neq x_1$ and $y_2 \neq y_1$.

Another trivial case for which $E[n_1[3]] \neq 0$ is when $x_2 = x_1$ and $(x_3 - x_1) + \cdots + (x_{n-k+2} - x_1) \neq 0$, or when $y_2 = y_1$ and $(y_3 - y_1) + \cdots + (y_{n-k+2} - y_1) \neq 0$.

**Code availability**

Software for this research is available in these in-text data citation references (Michalak, 2024b).

**Creator**: Michał Michalak. **Name of code**: 3GeoCombine2. **License**: GNU General Public License v3.0. **Contact address**: AGH University of Cracow, Faculty of Geology, Geophysics and Environmental Protection, Poland. E-mail: michalm@agh.edu.pl. **Year first available**: 2021. **Hardware required**: Celeron CPU or better. **Software required**: Code::Blocks. v. 20.03. **Program language**: C++. **Program size**: 482 KB. **How to access the source code**: Available at: https://doi.org/10.5281/zenodo.13974878 , https://github.com/michalmichalak997/3GeoCombine2/blob/main/README.md

**Data Availability**

Datasets for this research (input and processed data) are available in these intext data citation references (Michalak, 2024a).



**Author contribution**

MM devised the project, wrote the computer code and the manuscript, performed the computations, formal analysis
(formulation of theorems and the majority of proofs) and discussed the results. JM conducted the formal analysis (revision of existing proofs and conducting a sub-proof in Proposition 4) and revised the manuscript, PM discussed the results.

**Competing interests**

The authors declare that they have no conflict of interest.

**Acknowledgements**

This research project was supported by the program „Excellence initiative – research university" for the AGH University. We thank Dr Joshua Davis for providing suggestions about simplification of some parts of the reasoning. We thank Professor Christian Gerhards for the comments regarding the structure of the manuscript. An AGH student Natalia Kubańska is
acknowledged for providing the GEBCO data set presented in the Discussion. ChatGPT was used as a language editing tool. The research activities of the second author (JM) were co-financed by the funds granted under the Research Excellence Initiative of the University of Silesia in Katowice and by the University of Silesia Mathematics Department (Iterative Functional Equations and Real Analysis program).

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
