# Peer review of "Computational modelling and analytical validation of singular geometric effects in fault data using a combinatorial approach"

_EGUsphere, 2024_

## Author Comment (AC1)

In this file, there are responses to the Reviewer#1.

**Responses to the Reviewer#1**

We thank the Reviewer #1 for the review and recommendation. We've added the observed limitations by the Reviewer #1 to the manuscript.

To better observe communicating our response, we divided our responses into three categories: Agree/Clarification/Disagree.

1.

| Suggestion, Question, or Comment from the Reviewer#1 | Author's Response | Change in the Manuscript |
|---|---|---|
| The manuscript by Michalak et al. introduces a new computational model to predict fault geometry in data-sparse environments. I have the following concerns, which do not necessarily preclude acceptance of the manuscript:

1. The model is restricted to dip-slip faults w/out elevation uncertainties. | **Agree**

This is a correct general assessment of our work. | We've added information to the Abstract that we analyze dip-slip faults. |

2.

| Suggestion, Question, or Comment from the Reviewer#1 | Author's Response | Change in the Manuscript |
|---|---|---|
| 2. This technique does not differentiate between normal and reverse dip-slip faults. | **Agree**

This is correct. | We've added a sentence about this limitation to Discussion (5.2). |

3.

| Suggestion, Question, or Comment from the Reviewer#1 | Author's Response | Change in the Manuscript |
|---|---|---|
| 3. There are no real-world case studies to validate the model. | **Clarification**

In this study, we are more interested in mathematical relationships between points and directions, and synthetic data provide a more suitable environment for this type of analysis.

However, please note that we provided real-world data (GEBCO) which have some of the properties discussed (constant elevation difference). | None. |

4.

| Suggestion, Question, or Comment from the Reviewer#1 | Author's Response | Change in the Manuscript |
|---|---|---|
| 4. The use of Python would be more advantageous for the growing geomodeling community, especially since existing tools like GemPy have already established. | **Clarification**

Please note that one advantage of using C++ is its speed, as compared to Python. This may be important if the output resulting from the combinatorial program is big. But we agree that for the community, some links with the existing open-source software would be appreciated. | None. |

5.

| Suggestion, Question, or Comment from the Reviewer#1 | Author's Response | Change in the Manuscript |
|---|---|---|
| Despite these concerns, I believe the manuscript fits well with the scope of the journal, | Thank you for this encouraging note. | Not applicable. |

| and I would recommend it for publication. | | |
|---|---|---|

**Other changes:**

We had a bug in the code in relation to the sample circular dispersion (there was $m_2^2$ but it should be just $m_2$). The code and the tabulated results have been revised accordingly.

According to the request of the Editorial support (Mrs Daria Karpachova), we've made the background of selected figures (1, 3, 4 and 5) less dark.

---

## Author Comment (AC2)

In this file, there are responses to the Reviewer#2.

We thank the Reviewer#2 for a very thorough review with many insights. We agree with 80-90% of the comments. Regarding the 10-20%, we either disagree, or there is a misunderstanding, or divergence in preference of some mathematical presentations.

To better observe communicating our response, we divided our responses into three categories: Agree/Clarification/Disagree.

**Responses to the Reviewer#2**

1.

| Suggestion, Question, or Comment from the Reviewer#2 | Author's Response | Change in the Manuscript |
|---|---|---|
| The authors propose a method to statistically characterize a fault segment orientation in data-sparse environments. The method relies on a direct triangulation of a faulted horizon, and a statistical analysis of the dip direction of the set of triangles that can be formed using one triangle edge on one of the fault walls and all the triangulation vertices on the other wall. In addition, the authors provide mathematical evidences showing that - under some restrictive hypotheses – their statistical analysis yields exact / robust predictions of fault dip direction. Overall, this manuscript seems to be an improvement on the work of Michalak et al., 2021 (see reference in the paper) that aims at explaining and solving some of the counterintuitive results found by the authors. | **Agree**

This is a correct general assessment of our work.

In addition, we would like to observe that the use of the combinatorial algorithm does not need triangulation. | Not applicable. |

2.

| Suggestion, Question, or Comment from the Reviewer#2 | Author's Response | Change in the Manuscript |
|---|---|---|
| To me, the presented method is a useful tool to assess fault geometry in the absence of direct fault observations (it is entirely based on displaced horizon observations), and the mathematical details provide | **Agree**

We thank for the comments about the relevance of our mathematical insights. | Not applicable.

See the later comments and our responses. |

| Suggestion, Question, or Comment from the Reviewer#2 | Author's Response | Change in the Manuscript |
|---|---|---|
| interesting insights to understand why the method works and what its potential caveats are. As such, I consider it can be of interest to the audience of Solid Earth and deserves to be published. However, I also have several major concerns that would require revision of the manuscript. | | |

3.

| Suggestion, Question, or Comment from the Reviewer#2 | Author's Response | Change in the Manuscript |
|---|---|---|
| Note: despite the concerns listed below, I would like to acknowledge the effort made by the authors to provide all the necessary details and information necessary to understand their work and reproduce it. | **Agree**

We thank you for this assessment about reproducibility. | Not applicable. |

4.

| Suggestion, Question, or Comment from the Reviewer#2 | Author's Response | Change in the Manuscript |
|---|---|---|
| **General concerns**
Most of my concerns come from the fact that the presented method and the related mathematical "proofs" rely on two extremely restrictive assumptions:
● a globally horizontal horizon (i.e., only local variations/noise around a constant mean depth) | **Agree**/Clarification

Indeed, we acknowledge that we used restrictive assumptions. However, we are not certain whether they should be considered extreme. For example, a standard geostatistical method such as simple kriging also uses a known and constant expected value at any point of the domain (see the attached Screenshot from Wackernagel, 1995). We started with a simplified scenario and we believe that future work can relax these assumptions. | We've added some information about similarities with simple kriging assumption to Discussion. |

**Simple kriging**

Take the data locations $\mathbf{x}_\alpha$ and construct at each of the $n$ locations a random variable $Z(\mathbf{x}_\alpha)$. Take an additional location $\mathbf{x}_0$ and let $Z(\mathbf{x}_0)$ be a random variable at $\mathbf{x}_0$. Further assume that these random variables are a subset of a second-order stationary random function $Z(\mathbf{x})$ defined over a domain $\mathcal{D}$. By *second-order stationarity* we mean that the expectation and the covariance are both translation invariant over the domain, i.e. for a vector $\mathbf{h}$ linking any two points $\mathbf{x}$ and $\mathbf{x}+\mathbf{h}$ in the domain:

$$\mathrm{E}\big[ Z(\mathbf{x}+\mathbf{h}) \big] = \mathrm{E}\big[ Z(\mathbf{x}) \big]$$

$$\mathrm{cov}\big[ Z(\mathbf{x}+\mathbf{h}), Z(\mathbf{x}) \big] = C(\mathbf{h})$$

The expected value $\mathrm{E}\big[ Z(\mathbf{x}) \big] = m$ is the same at any point $\mathbf{x}$ of the domain. The covariance between any pair of locations depends only on the vector $\mathbf{h}$ which separates them.

Reference:

Wackernagel, H., (1995) Multivariate Geostatistics: An Introduction with Applications, Springer, p. 18.

5.

| Suggestion, Question, or Comment from the Reviewer#2 | Author's Response | Change in the Manuscript |
|---|---|---|
| • a vertical fault with one single segment (i.e., whatever the subset of the fault surface you consider, it will systematically have the same orientation) | **Clarification**

We are not sure if we understand this sentence correctly. Therefore, we provide two, potentially relevant, replies:

1. We agree that the assumption of a single, vertically oriented fault segment is restrictive. However, for faults with complex geometry, the task of assigning a single, consistent orientation to the entire surface becomes ill-posed. Our current approach focuses on a simplified scenario to isolate and validate the core methodology, with the intent to generalize to more complex fault | |

| | | |
|---|---|---|
| | geometries in future work.

2. In the manuscript, we didn't conduct an experiment with many fault segments. Therefore, we cannot judge about the statistical distribution of orientations for the case study with many fault segments. However, using the proven computational geometry principle, we would expect at least one significant difference with the case study with many fault segments. For example, for the case study with one segment the edge "e" in the below figure has no points to the right. But with two segments (see the below figure), we should expect two directions because there are points to the right. Therefore, the statistical distribution of directions will be different. | |

6.

| Suggestion, Question, or Comment from the Reviewer#2 | Author's Response | Change in the Manuscript |
|---|---|---|

| | | |
|---|---|---|
| Main concerns:
● I would talk about "mathematical evidences" rather than formal "mathematical proofs" throughout the manuscript (starting with the abstract), and only keep the term "Proof" in the appendices where it effectively corresponds to the proofs of mathematical propositions | Disagree/Clarification

We note that *"evidences"* is uncommon in the English language, as *"evidence"* is generally treated as an uncountable noun (https://www.ldoceonline.com/dictionary/evidence) .

We acknowledge that the use of "historical evidence" is justified but the phrase *"mathematical evidence"* could be considered vague and lead to misunderstandings.

That said, in most parts of the manuscript, our intention was to refer to formal logical arguments derived within a mathematical framework—thus, the term *"proofs"* seems more appropriate in those specific cases. In a more general context, we could speak about "mathematical reasoning" as the process conducted in the proofs. | We have revised the manuscript to use *"formal mathematical reasoning"* in more general contexts, such as in the abstract. |

7.

| Suggestion, Question, or Comment from the Reviewer#2 | Author's Response | Change in the Manuscript |
|---|---|---|
| ● Although the authors defend this choice of such ideal conditions for the sake of mathematical investigations, I would appreciate it if they discussed the expected results of the method in "not so ideal" conditions (e.g., dipping/folded horizons, normal/reverse faults, non-planar fault surfaces, ...). This would help to develop | **Clarification**

Please note that some of the requested topics (normal/reverse faults, dipping horizons and faults striking perpendicular to them) are already discussed in Michalak et al. 2021 (see a Screenshot with References below). To avoid redundancy, we prefer not to repeat those results here (see attached reference/screenshot).

Alternatively, we would like to focus and possibly expand on | We've added a sentence confirming that the current manuscript addresses a key limitation discussed in Michalak et al. (2021)— specifically, the issue of spatial distribution of points in relation to the boundary of the study area and the fault strike. |

| | | |
|---|---|---|
| the discussion which is otherwise short. | methodological challenges that we addressed since 2021 (Michalak et al. 2021):

1. a deeper understanding of the method - justification why it performs well by giving a statistical preference to edges parallel to the fault strike

 2. relevance for a popular geoscientific data set (GEBCO).

Please note that the provided data set is already less ideal than that presented in Michalak et al. 2021 (we've added errors to elevations). Future work should bring extensions to even less ideal data sets. | |

is influenced by both the fault and contact surface geometry.

The proposed methodology is limited to better calculating the fault strike but not the dip direction. This observation follows from the fact that the distribution of dip direction associated with triangles genetically related to a fault can be essentially attributed to both normal and reverse faults underlying this distribution. But in a case of a reverse fault (Fig. 4C) the dip direction of the fault would be opposite to that of the triangles. Therefore, to obtain the fault strike, 90° should be added to the calculated mean dip direction for the triangles' population. Theoretically, the dip direction of triangles could be consistent with the dip direction of a reverse fault, but this would require repeated layers in a borehole. However, this effect is equivalent to identifying a reverse fault, and our method is offered if no preferential effects of this kind are available. Although we investigated the directional variability of triangles genetically related to faults, this approach may not be sufficient for identifying faults striking perpendicular to the preferred dip direction, which is the case for homoclines (Michalak et al., 2019). Thus, distinguishing such faults from the preferred dip direction should also involve an analysis of the dip angle. Another limitation is that this method would make no difference between a fault and a ramp, which is the case for en echelon structures (Julio et al., 2015a, 2015b).

References:

Michalak, M. P., Kuzak, R., Gładki, P., Kulawik, A., & Ge, Y. (2021). Constraining uncertainty of fault orientation using a combinatorial algorithm. *Computers & Geosciences*, *154*, 104777.

8.

| Suggestion, Question, or Comment from the Reviewer#2 | Author's Response | Change in the Manuscript |
|---|---|---|
| | | |

| Suggestion, Question, or Comment from the Reviewer#1 | Author's Response | Change in the Manuscript |
|---|---|---|
| ● I would love to see an example of the application of the method on a real dataset. It could be applied on bathymetric data, as the authors already present it as one of the most direct use cases and discuss the pitfalls associated with such data | **Clarification**

We would like to clarify the intended purpose of the bathymetric (GEBCO) data used in this study. These data were not used to test the combinatorial algorithm for the analysis of fault orientation, as we currently lack evidence of any known fault structures in the seafloor region covered by the dataset. Therefore, applying our algorithm directly to such data would not be meaningful.

Instead, we used GEBCO data to highlight potential limitations in directional analyses (e.g., azimuth maps or circular histograms) when applied to triangulated bathymetric surfaces. For example, datasets with integer elevation values and a quasi-regular grid may produce identical azimuth values for multiple triangle edges. This broadens the relevance of our study beyond structural geology and tectonics, making it valuable to a wider geoscientific community interested in terrain or seafloor surface analysis. | None. |

9.

| Suggestion, Question, or Comment from the Reviewer#2 | Author's Response | Change in the Manuscript |
|---|---|---|
| ● A more detailed point: all the hypotheses are clearly stated in the proofs (in the appendices), but it is not so clear in the main body of the manuscript. Typically, I did not find the information that we assume a horizontal (constant Z) horizon. | **Clarification**

The assumption of constant elevation was indeed present in the caption of Fig. 3: *"The points on the surfaces of the hanging wall and footwall have constant elevation."* Additionally, this assumption was stated in Proposition 1: *"When the difference between* | We've added the assumption in the short summary of Proposition 1 (section 4.1). |

| | | |
|---|---|---|
| Please add a paragraph before stating the propositions, to clearly state all the hypotheses, as we can find in the appendices | *the hanging wall and the footwall is constant, then the following facts hold."*

 Given the direct link between the propositions and the panels in Fig. 3, we aimed to avoid redundancy.

 However, we acknowledge that this information could be inserted in section 4.1, in the "short summary" of Proposition 1. | |

10.

| Suggestion, Question, or Comment from the Reviewer#2 | Author's Response | Change in the Manuscript |
|---|---|---|
| Minor concerns
 • One point is still unclear to me after reading the manuscript: I feel like it is somehow mandatory to know in advance the position of the fault relatively to the data points to apply the method as presented here. This does not sound like a very realistic use case, and it seems quite straightforward to think about applying the method to try to identify unknown faults from horizon data. I would like the authors to clarify this point. | **Clarification**

 Indeed, the fault must be within the set of points used as input in the combinatorial algorithm.

 We agree that in many real-world cases, the location of faults may not be known in advance. In such situations, we suggest that a supervised fault detection method (e.g., Michalak et al., 2025, see reference below) be used first to identify potential fault locations based on horizon data. Once a candidate fault region is identified, the method proposed in this manuscript can be applied to infer its orientation.

 As of 8 May 2025, the manuscript on fault identification has been "accepted with corrections" in *Geoscientific Model Development*. | None. |

**References:**

Michalak, M., Gerhards, C., Menzel P., (2025) SubsurfaceBreaks v. 1.0: A supervised detection of fault-related structures on triangulated models of subsurface homoclinal interfaces, accepted with corrections in Geoscientific Model Development.

11.

| Suggestion, Question, or Comment from the Reviewer#2 | Author's Response | Change in the Manuscript |
|---|---|---|
| ● An important part of the "mathematical evidences" proposed by the authors relies on statistics ("the expectations of the coordinates", ...) whereas the method is presented for data-sparse studies. I agree that using the combinatorial strategy proposed by the authors provides more samples for computing statistics, but to me there remains a bias due to the limited number of "genetic triangles" edges. Again, it would be nice to have some additional discussion on this point | **Clarification**

This is an interesting and valuable comment. The use of statistical expectations in our method can indeed be better justified through analogy. Consider the classic coin tossing experiment: tossing a coin 10 times does not guarantee exactly 5 heads and 5 tails. However, as the number of tosses increases, the observed ratio tends to converge toward the expected 50/50 distribution.

Similarly, in our context, each experimental setup—e.g., drilling boreholes near a fault—can be viewed as a single realization. Although the geometry of the fault remains fixed, the spatial configuration of sampling points may vary. When this process is repeated many times (conceptually or through simulation), the statistical expectation of the reconstructed parameters converges toward the values derived in the deterministic framework, as shown in Appendix C.

This is analogous to the expected number of heads in a 10-coin toss experiment: while the actual outcome may be 4, 5, or 6, the expectation remains 5, which is the most rational estimate in the absence of further information. Likewise, the expected normal vector in our case represents the most consistent orientation expected across many realizations. In practice, a geologist may obtain a specific average (e.g., 4, 5, or 6 heads) from a single experiment, and this observed | None. |

average — although not exactly the expected value — should be accepted as the best available estimate under the constraints of the data. It reflects the statistical expectation derived from repeated trials or simulations. Importantly, the sample average is an **unbiased estimator** of the expected value — it does not systematically overestimate or underestimate it.

Comments from the annotated PDF.

12.

| Suggestion, Question, or Comment from the Reviewer#2 | Author's Response | Change in the Manuscript |
|---|---|---|
| Title: „strategy/approach?" | **Agree** | We've replaced "algorithm" with "approach" in the title. |

13.

| Suggestion, Question, or Comment from the Reviewer#2 | Author's Response | Change in the Manuscript |
|---|---|---|
| Line 24:
Not necessary? | **Agree** | We've deleted the last sentence from the abstract. |

14.

| Suggestion, Question, or Comment from the Reviewer#2 | Author's Response | Change in the Manuscript |
|---|---|---|
| Line 40:
After reading the manuscript, I understand the meaning, but when reading the first time I assumed it was about "reverse fault" or "inversion of the fault throw" | **Agree** | We've replaced "reversed" with "counterintuitive". |

15.

| Suggestion, Question, or Comment from the Reviewer#2 | Author's Response | Change in the Manuscript |
|---|---|---|
| Line 41 (regarding Fig. 4b): In Michalak et al, 2021 ? | **Clarification** No, we mean Fig. 4b in this manuscript (the turquoise arrow). | None. |

16.

| Suggestion, Question, or Comment from the Reviewer#2 | Author's Response | Change in the Manuscript |
|---|---|---|
| Line 66: References for cokriging-based modelling: Lajaunie et al., 1997 (https://doi.org/10.1007/BF02775087); Calcagno et al, 2008 (https://doi.org/10.1016/j.pepi.2008.06.013) | **Agree** | The references have been inserted in the manuscript. |

17.

| Suggestion, Question, or Comment from the Reviewer#2 | Author's Response | Change in the Manuscript |
|---|---|---|
| Line 77: Rather use "the algorithm described by Lipsky to generate all possible combinations of k elements among n". I do not think this is an officially named algorithm, as "the Metropolis-Hastings algorithm" can be | **Agree** | We've replaced "Lipski algorithm" with "combinatorial algorithm described by Lipski". |

18.

| Suggestion, Question, or Comment from the Reviewer#2 | Author's Response | Change in the Manuscript |
|---|---|---|
| Line 83: Please state here all the definitions of Michalak et al., 2021 you use (genetically related, ...), rather than in the appendices. It will help a lot the | **Agree/Clarification** We acknowledge that some of the definitions were already | We have revised and updated the relevant definitions, and moved them from the Appendix into the main body of the manuscript for clarity. We |

| | presented in the caption of Fig. 1. | have also modified the caption to Fig. 1 accordingly. |
| reader to understand, as the terminology is not standard at all | However, we now realize that certain definitions from Michalak et al. (2021) are no longer fully applicable in the current study. For example, "triangles genetically related to the fault" were previously defined as non-horizontal triangles. In this study, however, due to the introduction of elevation uncertainties, some non-horizontal triangles may no longer share the same geometric or genetic relationship with the fault—as all three vertices may lie on the same side. | |

19.

| Suggestion, Question, or Comment from the Reviewer#2 | Author's Response | Change in the Manuscript |
| --- | --- | --- |
| Lines 95-98: As you do not use it in the following, keep it concise and replace the paragraph by a single sentence stating that you cannot directly average vector coordinates | **Clarification**

The purpose of this paragraph is not to claim that vector coordinates cannot be averaged, but to emphasize that averaging only selected components of 3D vectors (e.g., X and Y) can lead to biased results. Omitting the Z component affects the relative contribution (or weight) of each vector in the resulting average.

If the goal is to analyze azimuths of 3D vectors, a better approach is to project the vectors onto the XY plane and normalize them. This removes the influence of dip on the mean azimuth. | We've modified a sentence in section 3.3 for clarity (the red phrase is new)

"Therefore, the directional ($X$ and $Y$) contribution of sub-horizontal triangles to the resultant vector will be relatively small compared to more inclined triangles (Fig. 2)." |

20.

| Suggestion, Question, or Comment from the Reviewer#2 | Author's Response | Change in the Manuscript |
|---|---|---|
| Lines 125-128:
I would remove this figure, this seems trivial enough to not need a graphical illustration | **Clarification/Disagree**

We would like to have this figure because it can be helpful in the following:

-the impression of "triviality" may be from a misunderstanding, as detailed in our response to comment #19

-the dimensionality reduction can also be relevant in terms of explaining apparent redundancy of orthogonality test (see our response to comment #44)

-immediate understanding of pitfalls regarding directional analysis of 3D and 2D data | None. |

21.

| Suggestion, Question, or Comment from the Reviewer#2 | Author's Response | Change in the Manuscript |
|---|---|---|
| Line 130:
Propositions? | **Agree** | We've replaced "formal results" with "propositions". |

22.

| Suggestion, Question, or Comment from the Reviewer#2 | Author's Response | Change in the Manuscript |
|---|---|---|
| Line 131:
As I said in the general comments, please add here a section to clearly state all the assumptions your mathematical proofs rely one | **Agree**
See our responses to comments #9 and #18. | Done. |

23.

| Suggestion, Question, | Author's Response | Change in the Manuscript |
|---|---|---|

| or Comment from the Reviewer#2 | | |
|---|---|---|
| Line 162: Again, propositions? | **Agree** | We've replaced "formal results" with "propositions". |

24.

| Suggestion, Question, or Comment from the Reviewer#2 | Author's Response | Change in the Manuscript |
|---|---|---|
| Line 181: cyan, not pink | **Agree** | Done. |

25.

| Suggestion, Question, or Comment from the Reviewer#2 | Author's Response | Change in the Manuscript |
|---|---|---|
| Line 205: I agree with Proposition 2 from a general point of view, but I see two difficulties: - from a formal / mathematical point of view: the dip direction of an horizontal plane is ill-defined (I assume...) - from a practical / implementation point of view: if your code considers that "an horizontal plane has a North-0 dip direction, then integrating the horizontal triangles in the statistical analysis will completely bias your result | **Clarification** We acknowledge that the dip direction of a horizontal plane is formally undefined, since the normal vector is vertical, i.e., [0, 0, 1], and this would indeed lead to division by zero in the atan2() function. However, we would like to clarify that adding the [0, 0, 1] vector to the vector sum in our statistical analysis does not affect the computed dip direction. For example, if the sum of normal vectors is [6, 7, 9], then adding [0, 0, 1] results in [6, 7, 10]. The dip direction remains unchanged, as it depends only on the x and y components of the resulting vector. Regarding the implementation issue: our GitHub repository includes three pieces of software, and in none of them do we assign any specific dip direction (e.g., North-0) to a horizontal plane or to the [0, 0, 1] vector. Nevertheless, if such a line was found, we kindly ask | None. |

| | the Reviewer to specify the file and line number.

As an alternative implementation strategy for 2D directional analysis, zero vectors ([0,0]) could be excluded, since they lack a defined direction and do not influence the statistical distribution of orientations. | |
|---|---|---|

26.

| Suggestion, Question, or Comment from the Reviewer#2 | Author's Response | Change in the Manuscript |
|---|---|---|
| Lines 210:
Would it change anything to the result if the error is spatially correlated? It would be a more realistic case. Maybe for the discussion? | **Clarification**

This is an interesting question. We agree that spatial correlation of errors could represent a more realistic scenario in certain contexts. However, since we did not perform simulations under this assumption, we are currently unable to assess its impact. | We've added a sentence about this limitation and possible development.

"Since our model assumes independence of errors across boreholes, another possible development would be to consider spatial correlation of errors. " |

27.

| Suggestion, Question, or Comment from the Reviewer#2 | Author's Response | Change in the Manuscript |
|---|---|---|
| Line 214:
See general comments: relying on expected values assumes to have a sufficient amont of data. I agree that "in data sparse contexts" it is still an interesting strategy to try to extract as much information as possible, but expected and values are not exactly the same | **Clarification**

See our response to comment #11. | None. |

28.

| Suggestion, Question, or Comment from the Reviewer#2 | Author's Response | Change in the Manuscript |
|---|---|---|
| Line 221:
Are we not supposed to work with directional statistics, rather than direct averaging of coordinates? | **Clarification**

Indeed, in directional statistics, there are several established approaches to computing the mean direction (see, e.g., Mardia & Jupp, *Directional Statistics*, Section 2.2). One of the standard methods assumes unit vectors and defines the mean direction as the direction of the resultant vector sum — which corresponds to computing the arithmetic mean of the Cartesian coordinates.

You can also have a look at the portion of Allmendinger et al. where direction cosines of orientation pairs (dip&dip direction) are regarded as Cartesian coordinates and summed to infer the mean orientation. | None. |

**2.1.1 Preliminaries and Notation**

Directions in the plane can be regarded as unit vectors $\mathbf{x}$, or equivalently as points on the unit circle (i.e. the circle of unit radius centred at the origin). There are two other useful ways of regarding such directions – as angles and as unit complex numbers. Choose an initial direction and an orientation for the unit circle. (This is equivalent to choosing an orthogonal coordinate system on the plane.) Then each point $\mathbf{x}$ on the circle can be represented by an angle $\theta$ or equivalently by a unit complex number $z$. These are related to $\mathbf{x}$ by

$$\mathbf{x} = (\cos\theta, \sin\theta)^T \quad \text{and} \quad z = e^{i\theta} = \cos\theta + i\sin\theta$$

(see Fig. 2.1).

**2.2 MEASURES OF LOCATION**

**2.2.1 The Mean Direction**

Suppose that we are given unit vectors $\mathbf{x}_1, \ldots, \mathbf{x}_n$ with corresponding angles $\theta_i$, $i = 1, \ldots, n$. The *mean direction* $\bar{\theta}$ of $\theta_1, \ldots, \theta_n$ is the direction of the resultant $\mathbf{x}_1 + \ldots + \mathbf{x}_n$ of $\mathbf{x}_1, \ldots, \mathbf{x}_n$. It is also the direction of the centre of mass $\bar{\mathbf{x}}$ of $\mathbf{x}_1, \ldots, \mathbf{x}_n$. Since the Cartesian coordinates of $\mathbf{x}_j$ are $(\cos\theta_j, \sin\theta_j)$ for $j = 1, \ldots, n$, the Cartesian coordinates of the centre of mass are $(\bar{C}, \bar{S})$, where

$$\bar{C} = \frac{1}{n}\sum_{j=1}^{n}\cos\theta_j, \qquad \bar{S} = \frac{1}{n}\sum_{j=1}^{n}\sin\theta_j. \qquad (2.2.1)$$

References: Mardia K., Jupp, P., Directional statistics, Wiley.

[Figure]

**Figure 2.7** Perspective diagram showing the relations between the trend and plunge angles and the direction cosines of the vector in the Cartesian coordinate system. Dark gray plane is the vertical plane in which the plunge is measured.

The solution to this problem uses vector addition and is shown graphically in Figure 2.11. Numerically, the steps are given below; for a computer program to solve this problem, see function `CalcMV` at the end of this section. The solution is illustrated with a real problem: Determine the mean vector of the following four lines, given as trend and plunge: 026, 31; 054, 22; 037, 39; and 012, 47.

1. Convert all of your orientation data into direction cosines.

| Trend and plunge | $\cos \alpha$ | $\cos \beta$ | $\cos \gamma$ |
|---|---|---|---|
| 026, 31 | 0.7704 | 0.3758 | 0.5150 |
| 054, 22 | 0.5450 | 0.7501 | 0.3746 |
| 037, 39 | 0.6207 | 0.4677 | 0.6293 |
| 012, 47 | 0.6671 | 0.1418 | 0.7314 |

2. Sum all of the individual components of the vectors, as in Equation 2.11. This will give you the resultant vector, **r**. If all the individual vectors have the same orientation, then the resultant vector will have a length that is equal to the number of vectors summed (in this example, $N = 4$); otherwise, it will always be less.

$$\sum \cos \alpha = 2.6032 \qquad \left(\sum \cos \alpha\right)^2 = 6.7767$$

$$\sum \cos \beta = 1.7354 \qquad \left(\sum \cos \beta\right)^2 = 3.0116$$

$$\sum \cos \gamma = 2.2503 \qquad \left(\sum \cos \gamma\right)^2 = 5.0639$$

Length of the resultant vector,

$$r = (6.7767 + 3.0116 + 5.0639)^{1/2} = 3.8539$$

3. Normalize the resultant vector by dividing each one of its components by the number of vectors summed together. The length of the normalized vector will always be less than or equal to 1. The closer it is to 1, the better the concentration.
   Note that $r = 0.9635$ indicates a reasonably strong preferred orientation.

$$\frac{\text{resultant length}}{N} = \frac{3.8539}{4} = 0.9635$$

4. Determine a unit vector, $\hat{\mathbf{m}}$, that is parallel to the resultant vector, **r**. To do this, calculate the magnitude of the resultant vector (or the normalized resultant vector) and then divide the components by the magnitude (Eqs. 2.3 and 2.4). These components will now be in direction cosines.

$$\hat{\mathbf{m}} = \left[\frac{2.6032}{3.8539} \quad \frac{1.7354}{3.8539} \quad \frac{2.2503}{3.8539}\right] = [0.6755 \quad 0.4503 \quad 0.5840]$$

5. Convert this final unit vector back to spherical coordinates.
   Trend and plunge of mean vector = 033.7°, 35.7°

'

References: Allmendinger, R. W., Cardozo, N., & Fisher, D. M. (2011). *Structural geology algorithms: Vectors and tensors*. Cambridge University Press.

29

| Suggestion, Question, or Comment from the Reviewer#2 | Author's Response | Change in the Manuscript |
|---|---|---|
| Line 222: | | |

| Suggestion, Question, or Comment from the Reviewer | Author's Response | Change in the Manuscript |
|---|---|---|
| When looking at the results on a stereonet: ok as you can distinguish the dip directions you are interested in "thanks to their dip" (about 90 degrees, when horizontal triangles are about 0). However, mathematically speaking the orientations of the horizntal triangles do not "simply cancel each other". Depending on the number of triagles on the footwall, it can completely change your overall distribution of dip directions, and drastically reduction the "signal-to-noise" ratio of the analysis | **Clarification**

We would like to clarify that in the original sentence (Line 222), we did not refer to the stereonet visualization, but to the effect of certain triangles on the **mean dip direction**, which is more evident in the **tabulated results** than in the stereonet itself.

Under the assumptions of Proposition 4, the X and Y components of the normal vectors associated with these triangles have zero expected value. Consequently, their contributions cancel out statistically in the mean, and do not affect the resulting average dip direction.

The situation is analogous to a coin toss experiment: although each realization may vary, the expected number of tails in 10 tosses remains 5. Similarly, even if individual normal vectors vary due to elevation uncertainty, the expected vector sum remains centered in the Z direction, [0, 0, 1], and does not bias the dip direction estimate. | None. |

30.

| Suggestion, Question, or Comment from the Reviewer#2 | Author's Response | Change in the Manuscript |
|---|---|---|
| Lines 249-252:
It would be interesting to compare the results with Michalak et al, 2021 to see the improvement | **Clarification**

The primary aim of our study was not to improve the results of Michalak et al. (2021), but rather to investigate how statistical estimates behave when elevation uncertainties are introduced. The purpose was to test the robustness of the method under such uncertainty, not to optimize or outperform existing approaches. | None. |

31.

| Suggestion, Question, or Comment from the Reviewer#2 | Author's Response | Change in the Manuscript |
|---|---|---|
| Lines 262-263: I did not get what orientations we are looking at in this figure | **Agree/Clarification** Indeed, it was not clear. We meant that they were the orientations of triangles as three-element subsets of the set of points. | We've extended the description about the data. |

32.

| Suggestion, Question, or Comment from the Reviewer#2 | Author's Response | Change in the Manuscript |
|---|---|---|
| Line 268: In (c), it "looks" like there are only punctual data, but if I well understood multiple points on the stereonet are colocalized. Comparing to (d): it becomes clear that these are actually clusters. Could it help adding density maps to the stereonet ? | **Clarification** Regarding the apparent colocalization of points: if the triangles share the same dip direction, their poles will indeed align along a single meridian in the polar projection. We chose not to apply density shading to the stereonets, as it could obscure this radial alignment pattern, which is important for interpreting dip direction consistency across realizations. | None. |

33.

| Suggestion, Question, or Comment from the Reviewer#2 | Author's Response | Change in the Manuscript |
|---|---|---|
| Line 276: Please remind here the ground truth values to help analyse the results | **Agree** | Done. |

34.

| Suggestion, Question, | Author's Response | Change in the Manuscript |
|---|---|---|

| | | |
|---|---|---|
| or Comment from the Reviewer#2 | | |
| Line 316:

No need to give example, or please cite an open-source alternative (e.g., QGIS) rather than a commercial software | **Agree** | We've deleted the example. |

35.

| Suggestion, Question, or Comment from the Reviewer#2 | Author's Response | Change in the Manuscript |
|---|---|---|
| Line 347:
This still remains a very idealized case, given the other assumptions | **Clarification**

We note that in this context „idealized" can be equivalent to „rounded": we demonstrated an example GEBCO data set which has rounded (integer) values which locally looks as the data points from Fig. 5A. | None. |

36.

| Suggestion, Question, or Comment from the Reviewer#2 | Author's Response | Change in the Manuscript |
|---|---|---|
| Line 351:
To generic, data-based triangulation instead? You can do TIN-based modelling without using your data as a direct support. See e.g., Caumon et al, 2009 (https://doi.org/10.1007/s11004-009-9244-2) among many others | **Clarification**

We meant that directional analyses of TIN data should take into account the propositions obtained in this study to better understand the directional distribution. | We've modified the bullet point. |

37.

| Suggestion, Question, or Comment from the Reviewer#2 | Author's Response | Change in the Manuscript |
|---|---|---|
| Line 394:
Remove the double negation for clarity | **Clarification** | We've moved the definitions from the Appendix to section 3.2. |

| | Please note that due to update of the definitions, this definition no longer exists. | See also our response to comment #18. |

38.

| Suggestion, Question, or Comment from the Reviewer#2 | Author's Response | Change in the Manuscript |
|---|---|---|
| Line 407:
Though obvious, introduce notations: given 3 vertices s = (s1, s2), ... | **Agree** | Done. |

39.

| Suggestion, Question, or Comment from the Reviewer#2 | Author's Response | Change in the Manuscript |
|---|---|---|
| Line 409:
The oriented vector tu. | **Agree** | Done. |

40.

| Suggestion, Question, or Comment from the Reviewer#2 | Author's Response | Change in the Manuscript |
|---|---|---|
| Line 414:
Already said, but I did not find this important assumption sooner in the text | **Clarification**

See our response to comment #9. | See our response to comment #9. |

41.

| Suggestion, Question, or Comment from the Reviewer#2 | Author's Response | Change in the Manuscript |
|---|---|---|
| Line 418:
For all equations: check SE guidelines for writing equations. I am not familiar with them, but I assume most of bold characters should be replaced by "normal" ones. Bold is usually for vectors / matrices, not for scalar values | **Agree/Clarification**

The SE guidelines say:

"Matrices are printed in boldface, and vectors in boldface italics."

However, I am not sure if all papers adopt this visualization. For example, the Sanan et al. paper | We've removed the bold style from scalar elements in vectors and matrices. |

| | (https://doi.org/10.5194/se-11-2031-2020 ) doesn't print matrices in bold. Nevertheless, you are probably right that matrix elements shouldn't be printed in bold. | |

42.

| Suggestion, Question, or Comment from the Reviewer#2 | Author's Response | Change in the Manuscript |
|---|---|---|
| Line 422:

To keep the manuscript concise, directly state "using the cross product, we compute the normal vectr..." and put the results of equations 13-15, without the intermediate steps. The computations are simple enough for the reader to be able to reproduce them on its own | **Clarification**

We agree that the computations are simple.

However, we would prefer to have the intermediate steps because it is more clear that we calculate the cross product for 3D vectors rather than 2D vectors (see also our response to comment #44). | None. |

43.

| Suggestion, Question, or Comment from the Reviewer#2 | Author's Response | Change in the Manuscript |
|---|---|---|
| Line 450:
Remove this sentence | **Agree** | Done. |

44.

| Suggestion, Question, or Comment from the Reviewer#2 | Author's Response | Change in the Manuscript |
|---|---|---|
| Line 455:

No need, this is the definition of the cross-product!
Its result is orthogonal to the "input" vectors | **Clarification**

Please note that the cross-product was done for 3D vectors, while the "orthogonality/perpendicularity" test was conducted using dot product for 2D vectors. | None. |

45.

| Suggestion, Question, or Comment from the Reviewer#2 | Author's Response | Change in the Manuscript |
|---|---|---|
| Line 455:

orthogonal | **Agree**

We think that both versions are good in this context, but we've replaced "perpendicular" with "orthogonal" | Done. |

46.

| Suggestion, Question, or Comment from the Reviewer#2 | Author's Response | Change in the Manuscript |
|---|---|---|
| Line 491:

Again, to keep it concise, directly state the result of equations 24-26 | **Clarification**

See our response to the comment #42. | None. |

47.

| Suggestion, Question, or Comment from the Reviewer#2 | Author's Response | Change in the Manuscript |
|---|---|---|
| Line 551:

Again | **Clarification**

See our response to the comment #42. | None. |

**Other changes:**

We had a bug in the code in relation to the sample circular dispersion (there was $m_2^2$ but it should be just $m_2$). The code and the tabulated results have been revised accordingly.

According to the request of the Editorial support (Mrs Daria Karpachova), we've made the background of selected figures (1, 3, 4 and 5) less dark.

---

## Author Comment (AC3)

In this file, there are responses to the Reviewer#3 (Giacomo Medici, community comment).

We thank the Reviewer#3 for additional efforts and comments leading to improvement of figures.

**Responses to the Reviewer#3 (Community comment, Giacomo Medici)**

1.

| Suggestion, Question, or Comment from the Reviewer#3 (community comment) | Author's Response | Change in the Manuscript |
|---|---|---|
| General comments

Very good geo-modelling research with a focus on representation of fault geometries. Please, follow my specific comments to improve the manuscript. | Thank you for your assessment. | Not applicable. |

2.

| Suggestion, Question, or Comment from the Reviewer#3 (community comment) | Author's Response | Change in the Manuscript |
|---|---|---|
| Specific comments

Line 16. "Geometrical" better than "directional" for an abstract. | **Agree** | Done. |

3.

| Suggestion, Question, or Comment from the Reviewer#3 (community comment) | Author's Response | Change in the Manuscript |
|---|---|---|
| Line 32. Add other applications in the growning fields of geo-sciences of $CO_2$ storage and geothermal energy. Please, insert the following references | **Agree/Clarification**

We believe that the paper (doi: 10.3389/feart.2023.1328397) is relevant because, as a review article, it summarizes | We've added new citations in relation to geothermal energy and $CO_2$ storage. |

| for the importance of faults in these two geo-energy fields:

Geothermal energy: Medici, G., Ling, F., Shang, J. 2023. Review of discrete fracture network characterization for geothermal energy extraction. Frontiers in Earth Science, 11, 1328397.

CO2 storage: Nicol, A., Seebeck, H., Field, B., McNamara, D., Childs, C., Craig, J., Rolland, A. 2017. Fault permeability and CO2 storage. Energy Procedia, 114, 3229-3236. | experimental techniques to characterize geometrical properties of subsurface aquifers which is relevant for geothermal applications. However, we would also need to cite at least one original article (https://doi.org/10.1016/j.geothermics.2022.102523). Regarding the second paper (doi: 10.1016/j.egypro.2017.03.1454), we believe that it is a conference paper or an extended abstract. Therefore, we decided to cite a full original paper instead (https://doi.org/10.1016/j.ijggc.2019.06.013). | |

4.

| **Suggestion, Question, or Comment from the Reviewer#3 (community comment)** | **Author's Response** | **Change in the Manuscript** |
|---|---|---|
| Line 50. Clearly state the 3 to 4 specific objectives of your geo-modelling research by using numbers (e.g., i, ii, and iii). | **Agree/Clarification**

Honestly, we have rather two main objectives: analyzing case studies for data with and without uncertainties. Of course, we could add a third bullet point with the analysis of GEBCO data, but this is rather a discussion of implications of the previous analyses. | We've highlighted two bullet points. |

5.

| Suggestion, Question, or Comment from the Reviewer#3 (community comment) | Author's Response | Change in the Manuscript |
|---|---|---|
| Page 6. I can see several equations without numbers associated with. | **Clarification**

Some minor formulas don't have numbers. If the Editor will require additional numbers, we will introduce them. | None, as of now, but can be added upon the Editor's request.. |

6.

| Suggestion, Question, or Comment from the Reviewer#3 (community comment) | Author's Response | Change in the Manuscript |
|---|---|---|
| Lines 281-294. This part of the discussion shows paucity of literature. I suggest to back-up your statements with supporting literature. | **Agree** | We've added three references to the section 5.1: two textbooks as a support for general definitions and one historical paper. |

7.

| Suggestion, Question, or Comment from the Reviewer#3 (community comment) | Author's Response | Change in the Manuscript |
|---|---|---|
| Line 362. Add a "take home message" for the researchers working in the field. | **Clarification**

We are not sure which field is the Reviewer referring to (structural geologists, bathymetric geomorphologists). The conclusions can be mostly useful for researchers working explicitly with triangulated surfaces or implicitly: in the form of performing directional analyses of surfaces (azimuth, | We've added a general statement before the bullet points.

We've improved the first bullet point of the Conclusion because it was not very clear. |

| | | |
|---|---|---|
| | maps) that use triangulations under the hood. | |

8.

| Suggestion, Question, or Comment from the Reviewer#3 (community comment) | Author's Response | Change in the Manuscript |
|---|---|---|
| Figures and tables

Figure 3. You can make the four diagrams closer, gain space and enlarge the overall image. The four blocks are difficult to analyse. | **Agree** | We've reduced the space between four diagrams and enlarged the image.

Moreover, we've made the background less dark according to the request of the Editorial support. |

9.

| Suggestion, Question, or Comment from the Reviewer#3 (community comment) | Author's Response | Change in the Manuscript |
|---|---|---|
| Figure 4c. This is a conceptually different image. It should represent a separate Figure 5. | **Clarification/Disagree**

Although Figure 4c might appear conceptually different at first glance, it is intentionally placed as part of Figure 4 because it directly corresponds to the triangle orientations shown above. Separating it into a new figure would break this link and make it harder for the reader to associate the triangle geometry with the resulting directional patterns. | None. |

10.

| Suggestion, Question, | Author's Response | Change in the Manuscript |
|---|---|---|

| Suggestion, Question, or Comment from the Reviewer#3 (community comment) | | |
|---|---|---|
| Figure 6c and d. Same issue here. These are very different images. They should represent a separate figure. | **Clarification**

See our response to comment #9.

However, we realize that the caption did not include all information to see the correspondence. | We've improved the caption to Fig. 6 to show the direct correspondence. |

11.

| Suggestion, Question, or Comment from the Reviewer#3 (community comment) | Author's Response | Change in the Manuscript |
|---|---|---|
| Figure 7c. Improve the graphical resolution of the Figure 7c which is a stereonet. | **Agree** | Done. |

**Other changes:**

We had a bug in the code in relation to the sample circular dispersion (there was $m_2^2$ but it should be just $m_2$). The code and the tabulated results have been revised accordingly.

According to the request of the Editorial support (Mrs Daria Karpachova), we've made the background of selected figures (1, 3, 4 and 5) less dark.

---

## Author Response (AR2)

In this file, there are responses to the Reviewer#2 (Anonymous Referee) and to the Associate Editor.

We thank the Reviewer#2 for the analysis of our responses and additional comments.

We thank the Associate Editor for continuous efforts to secure the reviews and for additional comments.

To better observe communicating our response, we divided our responses into three categories: Agree/Clarification/Disagree.

**Responses to the Reviewer#2**

1.

| Suggestion, Question, or Comment from the Reviewer#2 | Author's Response | Change in the Manuscript |
|---|---|---|
| I am generally satisfied with the authors' response. However, I remain concerned about the size and content of the manuscript. Notably, a significant portion of the manuscript consists of reminders from the 2021 paper by Michalak et al. This includes the description of the combinatorial approach, the related algorithm, and the statistical analysis (including the differences between 2D and 3D). The original content of the manuscript is limited to the four mathematical propositions and their proofs (to the authors' credit, this is clearly stated in the introduction).

This contribution is interesting but seems relatively brief to justify a standalone publication. I agree with the authors' response that most of the topics proposed in the different reviews for additional discussions (e.g., the effect of normal/reverse faults) would partly change the scope of the paper and that some of these topics are already discussed in | **Clarification**

We respectfully emphasize that the original contribution of the manuscript extends beyond the formal mathematical propositions. In particular, we:

1. introduce and analyze the effect of **elevation uncertainty** on the statistical behavior of the method — a feature not present in Michalak et al. (2021).

2. Moreover, we broaden the applicability of the method by discussing its relevance to real-world geoscientific datasets such as **GEBCO bathymetry**, highlighting previously overlooked artifacts (e.g., azimuth clustering).

We would also like to note that in the initial review, concerns about the originality or scope of the contribution were not listed among the reviewer's "main concerns", and we addressed all major and detailed points raised at that stage. | None. |

| | | |
|---|---|---|
| the 2021 paper. While I can hardly suggest other propositions, I still believe the paper would truly benefit from some additions… | | |

2.

| Suggestion, Question, or Comment from the Reviewer#2 | Author's Response | Change in the Manuscript |
|---|---|---|
| Detailed remarks :

 - L. 45 : I would remove the sentence "We propose a robust framework for predicting fault geometry in data-limited scenarios" as it could be misleading and interpreted as "the approach described in this paper is new" | **Agree** | Done. |

3.

| Suggestion, Question, or Comment from the Reviewer#2 | Author's Response | Change in the Manuscript |
|---|---|---|
| - Following up the discussion on the introduction of footwall triangles in the statistical analysis (cf. propositions 2 and 4): I agree with the authors that they statistically do not affect the computed mean dip direction (the way it is computed is clearer to me thanks to the authors' response). However, they have an impact on the circular standard error and as such they artificially narrow the N % confidence intervals that are deduced from it (uncertainty quantification, etc.) | **Clarification**

 In our implementation, **horizontal and vertical triangles are explicitly excluded from the statistical analysis**.

 Prior to computing directional averages and measures of dispersion, we remove all horizontal and vertical triangles.

 **As a result, horizontal triangles do not contribute to the denominator in the formula for the sample circular dispersion**, nor do they affect the confidence interval estimates.

 This filtering is presented in the code attached below. It seems | None. |

| | that the Referee noticed this effect (in the #4 comment). | |
|---|---|---|

```
29
30    #Combinatorial results
31    surface <- read.table(".txt", header=TRUE, sep = ";", dec=".")
32    nrow(surface)
33
34    surface<-dplyr::filter(surface, DOC<1)#deleting collinear configurations
35    surface<-dplyr::filter(surface, Z_N<1)#deleting horizontal samples
36    surface<-dplyr::filter(surface, Z_N>0)#deleting vertical samples
37    N=nrow(surface)    #number of valid samples
38    N
39
```

4.

| Suggestion, Question, or Comment from the Reviewer#2 | Author's Response | Change in the Manuscript |
|---|---|---|
| - About Table 1: There is not the same number of observations between Fig.5 (a, c) and (b, d), and between Fig.6 (a, c) and (b, d). I assume that the footwall triangles are added (only) in the (b, d) figures. It could be interesting to also integrate these triangles in the (a, c) figures, so we can compare the impact of elevation errors on the confidence intervals. Maybe having 2 cases for (a, c) figures (with and without these triangles) would be nice, so we can also have a measure of their impact on the confidence intervals. | **Clarification**

We would like to clarify that, in all cases, **horizontal triangles are explicitly excluded from the statistical analysis**. Including such triangles would, in fact, be problematic — particularly in the 2D case — since a [0, 0] vector **cannot be assigned a direction**, and therefore cannot be used in directional statistics.

We also note what may appear as two opposing ideas in the reviewer's comments: one suggests that horizontal triangles may artificially narrow confidence intervals (which would support their removal), while the other proposes including them in more cases to assess their influence. Given our filtering approach, this issue does not arise in our analysis. | None. |

**Responses to the Associate Editor**

1.

| Suggestion, Question, or Comment from the Associate Editor | Author's Response | Change in the Manuscript |
|---|---|---|
| - L. 45 : I would remove the sentence "We propose a robust framework for predicting fault geometry in data-limited scenarios" as it could be misleading and interpreted as "the approach described in this paper is new" | **Agree** | Done. |

2.

| Suggestion, Question, or Comment from the Associate Editor | Author's Response | Change in the Manuscript |
|---|---|---|
| In addition, there are a few points of clarity that I would like to be addressed:

1. Why is the colormap in Figure 6a and 6b different? Could they be the same? | **Clarification**

The difference in colormap arises because panel (b) includes **random elevation perturbations**, whereas panel (a) shows an **idealized configuration with constant elevation steps**.
To enhance visual contrast, each panel uses an **independent color scale**, tailored to the actual range of elevation values in that panel. In panel (b), the elevation values — due to added noise — are concentrated near both **the upper and lower ends** of the scale, resulting in stronger color variation.

Using a common color scale across both panels would reduce the visibility of these subtle but important differences in elevation distribution | None |

|  |  |  |
| --- | --- | --- |
|  |  |  |

3.

| Suggestion, Question, or Comment from the Associate Editor | Author's Response | Change in the Manuscript |
| --- | --- | --- |
| 2. Line 305 - could we change "We note that we addressed the problem posed by (Michalak et al., 2021) in Discussion: the issue of spatial distribution of points in relation to the boundary of the study area and the fault strike (Figs. 6a, 6b)." to

"Previous work identified an issue of the spatial distribution of points in relation to the boundary of the study area and the fault strike (Michalak et al., 2021). However, the work here addressed this problem (Figure 6a, 6b)."

Or, could you describe in more detail the issue (and how you have overcome it) rather than referring to the Discussion section of a previous paper. | **Agree** | We revised the manuscript, according to the suggestion. |

Other changes:

- We've added the word „horizontal" in Observation 1, because this observation relates to the simplest scenario
- In Part B of Proposition 1, we specified that we mean a 2D variant for the projected vector
- We've applied italics/bold style, where it is required. We've applied vector style (bold and italics) for „edges"

---

## Author Response (AR3)

In this file, there are responses to the Associate Editor.

We thank the Associate Editor for constructive suggestions.

To better observe communicating our response, we divided our responses into three categories: Agree/Clarification/Disagree.

**Responses to the Associate Editor**

1.

| Suggestion, Question, or Comment from the Associate Editor | Author's Response | Change in the Manuscript |
|---|---|---|
| 1. In point one of the revision you state very clearly the differences between Michalak et al. (2021) and this new paper, but do not change any text. I think given the concern of the reviewer it would be sufficient to add your described differences to the Introduction:

Line 44: "This paper builds on this work by providing formal mathematical reasoning that a combinatorial algorithm can reduce epistemic uncertainty in sparse environments. Specifically, here we introduce and analyze the effect of elevation uncertainty on the statistical behavior of the method and present formal analysis of two scenarios:"

Line 48: "Following the formal analysis, the work further extends from Michalak et al. (2021) by discussing its relevance to real-world geoscientific datasets. Here, we demonstrate the consequences of these theoretical results in the analysis of 2D and 3D (Fig. 2) directional data derived from topographic grids, which typically consist of points with approximate elevations—commonly observed in bathymetric datasets (Gridded Bathymetry Data, 2024)." | **Agree** | Done. |

2.

| Suggestion, Question, or Comment from the Associate Editor | Author's Response | Change in the Manuscript |
|---|---|---|
| On my point of Figure 6 - I would like to see what the figure looks like without editing the colormap to highlight what you want to show. By having the colourmaps so very different, it is not possible to compare the two panels (which is what you want to do) as they are presented with different colours. Could you please show the data for Fig 6b using the same colourmap. If that image looks exactly the same as Fig 6a, you could produce a figure of % error instead of elevation for panel b? I am happy to hear your thoughts on this. | **Clarification**

Yes, it is possible to show the data for Fig. 6b using the same color scale.
We have attached two ParaView screenshots below (one with the default color scale and one with an adjusted scale) for comparison. Please note that we applied a side view here (whereas the manuscript uses a top view) to better illustrate the issue.
These screenshots show that the data do not have identical elevation. However, the default color scale (b) does not effectively reveal these differences, whereas the adjusted scale makes them more apparent.
This distinction is even more critical in the top view, where elevation differences cannot be detected based on point position alone. Therefore, an appropriately adjusted color palette is essential in the top view.
We are not ParaView experts, so there may be better ways to differentiate elevation visually, but to the best of our knowledge, this is the most effective approach available.
We prefer to present elevation values rather than "% error" because the manuscript occasionally refers to "identical elevation," and keeping the figure in terms of elevation ensures conceptual consistency. | We have added a clarification to the caption:

„Due to the subtle elevation differences, a top view was chosen to better illustrate the spatial layout of the points in map view; in such a projection, elevation variations are not visually evident, hence a carefully adjusted color scale (b) is crucial for interpretation." |

**Data with elevation uncertainties (Fig 6b, side view – „scalar" denotes elevation):**

  a)  **with adjusted color scale**
  b)  **without adjusted color scale**

**a)**

[Figure]

**b)**

[Figure]

---

## Author Response (AR4)

In this file, there are responses to the Associate Editor.

We thank the Associate Editor for a quick reply.

To better observe communicating our response, we divided our responses into three categories: Agree/Clarification/Disagree.

**Responses to the Associate Editor**

1.

| Suggestion, Question, or Comment from the Associate Editor | Author's Response | Change in the Manuscript |
|---|---|---|
| Thank you for your prompt response. For Figure 6 - I'm still unable to assess the point you make with the figure projection you provide in the response (e.g., map view).

Can you provide a version of Figure 6 that has panel 'a' next to a new panel 'b' which has the same colourmap as panel 'a' (e.g., top down view, scale from 1.1 to -1.1)? Then we can compare whether panel 'a' and panel 'b' show any differences in colour variation (as you mention). | **Clarification**

Yes, it is possible to provide a version of Figure 6 with two panels that have the same colourmap (see the screenshot attached below).

In our opinion, panel (b) looks very similar to panel (a).

Please note that due to compression the embedded file here is not of the best quality and the corresponding author has sent an e-mail with the figure – it seems that the system does not offer to provide any additional files except the manuscript and responses.

Regarding potential questions about ranges: We are not entirely certain why the displayed ranges differ, but we suspect it may be due to different ParaView versions used during figure generation. | Please note that we haven't changed the figure in the manuscript. However, we can do this after the comparison and a potential request. |

An additional suggestion:

In case of more granular analyses, please note that the files for visualization (PVSM and VTK) are available in our Zenodo repository. For example, if you'd like to open the panel (b) in ParaView, please visit the below attached Zenodo repository. In ParaView, click „Load state" and choose the „fault_oblique_regular_grid_error_with.pvsm" file. Then, provide the path for the file „fault_oblique_regular_grid_error_with_Cloud.vtk".

**References:**

Michalak, M.: Computational modeling and analytical validation of singular geometric effects in fault data using a combinatorial approach - Input and processed data, https://doi.org/10.5281/zenodo.13986509, 2024a

2.

| Suggestion, Question, or Comment from the Associate Editor | Author's Response | Change in the Manuscript |
|---|---|---|
| We do not need the previous clarification to the caption of: "„Due to the subtle elevation differences, a top view was chosen to better illustrate the spatial layout of the points in map view; in such a projection, elevation variations are not | **Agree/Clarification**

We agree that the term *"map view"* is not discussed in the manuscript.

However, with this clarification we aimed to emphasize the | We've deleted the clarification from the manuscript. |

| | | |
|---|---|---|
| visually evident, hence a carefully adjusted color scale (b) is crucial for interpretation." We don't discuss map view in the paper and as such it would be confusing to the reader. | *layout* — that is, the spatial arrangement of points relative to the fault strike and the boundaries of the study area.

Nevertheless, we have removed this sentence from the caption and will wait for any further suggestions regarding the wording. | |

---

## Author Response (AR5)

In this file, there are responses to the Associate Editor.

We thank the Associate Editor for a quick reply.

To better observe communicating our response, we divided our responses into three categories: Agree/Clarification/Disagree.

**Responses to the Associate Editor**

1.

| Suggestion, Question, or Comment from the Associate Editor | Author's Response | Change in the Manuscript |
|---|---|---|
| Thanks so much for the reply - and for taking the time to explore how best to accurately represent the data in Figure 6.

Based on the figures you've sent, I think the 1.2 to -1.2 with the grey background shows the subtle changes in the elevation, and as a result it is more suitable (and accurate) then the original version. The reasons for this are that:

1. your elevation changes are small, so there is no need to overstate the changes (actually, it is better for the results of the paper to highlight that the changes are only slight). The original image overstated the changes in the data.

2. The data range is fully captured with the 1.2 to -1.2. As a result, there is no topping-out of the data, which is the most accurate way to show the information.

3. The figures can now be easily and accurately compared as they are now on the same colourmap and scale. By having two different colourmaps the original work (unintentionally) distorted the data to show more changes than there actually are. The data change in Figure 6 is subtle and that's ok - let's present the data without distortion where you can see | **Agree** | We have updated Fig. 6 and the graphical abstract, accordingly. |

| the slight changes. I think this is achieved with the 1.2 and -1.2 figure here with the grey background.

My recommendation for revision is to update the Figure 6 to 1.2 to -1.2 scale for both (and to not forget to add the pink polygon) - I believe the main figure after the abstract should be updated also.

Thanks for your work on this, and the push for clear representation of the data. | | |
|---|---|---|

The updated figure: